# Mouse Models of Human Pathogenic Variants of *TBC1D24* Associated with Non-Syndromic Deafness DFNB86 and DFNA65 and Syndromes Involving Deafness

**DOI:** 10.3390/genes11101122

**Published:** 2020-09-24

**Authors:** Risa Tona, Ivan A. Lopez, Cristina Fenollar-Ferrer, Rabia Faridi, Claudio Anselmi, Asma A. Khan, Mohsin Shahzad, Robert J. Morell, Shoujun Gu, Michael Hoa, Lijin Dong, Akira Ishiyama, Inna A. Belyantseva, Sheikh Riazuddin, Thomas B. Friedman

**Affiliations:** 1Laboratory of Molecular Genetics, National Institute on Deafness and Other Communication Disorders, Porter Neuroscience Research Center, National Institutes of Health, Bethesda, MD 20892, USA; risa.tona@nih.gov (R.T.); cristina.fenollarferrer@nih.gov (C.F.-F.); rabia.faridi@nih.gov (R.F.); belyants@nidcd.nih.gov (I.A.B.); 2The NIDCD National Temporal Laboratory at UCLA, Department of Head and Neck Surgery, David Geffen School of Medicine at UCLA, Los Angeles, CA 90095, USA; ilopez@ucla.edu (I.A.L.); ishiyama@ucla.edu (A.I.); 3Laboratory of Molecular & Cellular Neurobiology, National Institute of Mental Health, National Institutes of Health, Bethesda, MD 20892, USA; 4Research Center for Genetic Medicine, Children’s National Hospital, Washington, DC 20010, USA; canselmi@childrensnational.org; 5Department of Genomics and Precision Medicine, The George Washington University, Washington, DC 20052, USA; 6National Centre of Excellence in Molecular Biology, University of the Punjab Lahore, Lahore 53700, Pakistan; asmaalikan@gmail.com; 7Department of Molecular Biology, Shaheed Zulfiqar Ali Bhutto Medical University, Islamabad 44080, Pakistan; mohsinshahzad@szabmu.edu.pk (M.S.); riazuddin@aimrc.org (S.R.); 8Jinnah Burn and Reconstructive Surgery Center, Allama Iqbal Medical Research Center, Jinnah Hospital, University of Health Sciences, Lahore 54550, Pakistan; 9Genomics and Computational Biology Core, National Institute on Deafness and Other Communication Disorders, NIH, Bethesda, MD 20892, USA; morellr@nidcd.nih.gov; 10Auditory Development and Restoration Program, National Institute on Deafness and Other Communication Disorders, National Institutes of Health, Bethesda, MD 20892, USA; shoujun.gu@nih.gov (S.G.); michael.hoa@nih.gov (M.H.); 11Genetic Engineering Core, National Eye Institute, National Institutes of Health, Bethesda, MD 20892, USA; dongl@nei.nih.gov

**Keywords:** *Tbc1d24* mouse models, hearing loss, DFNB86, DFNA65, DOORS, syndromic deafness, human temporal bone

## Abstract

Human pathogenic variants of *TBC1D24* are associated with clinically heterogeneous phenotypes, including recessive nonsyndromic deafness DFNB86, dominant nonsyndromic deafness DFNA65, seizure accompanied by deafness, a variety of isolated seizure phenotypes and DOORS syndrome, characterized by deafness, onychodystrophy, osteodystrophy, intellectual disability and seizures. Thirty-five pathogenic variants of human *TBC1D24* associated with deafness have been reported. However, functions of TBC1D24 in the inner ear and the pathophysiology of TBC1D24-related deafness are unknown. In this study, a novel splice-site variant of *TBC1D24* c.965 + 1G > A in compound heterozygosity with c.641G > A p.(Arg214His) was found to be segregating in a Pakistani family. Affected individuals exhibited, either a deafness-seizure syndrome or nonsyndromic deafness. In human temporal bones, TBC1D24 immunolocalized in hair cells and spiral ganglion neurons, whereas in mouse cochlea, *Tbc1d24* expression was detected only in spiral ganglion neurons. We engineered mouse models of *DFNB86* p.(Asp70Tyr) and *DFNA65* p.(Ser178Leu) nonsyndromic deafness and syndromic forms of deafness p.(His336Glnfs*12) that have the same pathogenic variants that were reported for human *TBC1D24*. Unexpectedly, no auditory dysfunction was detected in *Tbc1d24* mutant mice, although homozygosity for some of the variants caused seizures or lethality. We provide some insightful supporting data to explain the phenotypic differences resulting from equivalent pathogenic variants of mouse *Tbc1d24* and human *TBC1D24*.

## 1. Introduction

Sensorineural hearing loss is a heterogenous disorder with a great variety of etiologies, including inherited and de novo pathogenic variants, infections, ototoxic drugs, and aging [1,2]. Variants of hundreds of different human genes are associated with deafness affecting approximately 0.2% of newborns [3]. Non-syndromic hearing loss transmitted as an autosomal recessive trait is often congenital, while transmission as an autosomal dominant trait often results in a progressive loss of hearing [4]. Approximately one-third of individuals with an inherited hearing loss have additional impairments in other organs [5], and of the more than 400 reported syndromic forms of hearing loss, many are not yet well-understood at the molecular level [6]. Moreover, different pathogenic variants of a gene may be associated with dominant or recessive modes of inheritance, and the phenotypic outcome may be syndromic or non-syndromic. For example, some variants of *MYO7A* are associated with dominant (*DFNA11*) or recessive (*DFNB2*) inheritance of non-syndromic deafness, while the majority of *MYO7A* pathogenic variants cause Usher syndrome (USH1B) [7,8,9].

Another example is pathogenic variants of *TBC1D24* that are associated with non-syndromic hearing loss, segregating as a recessive (*DFNB86*) [10] or a dominant trait (*DFNA65*) [11,12]. Variants of *TBC1D24* are also associated with syndromic hearing loss, which includes hearing loss with seizures, or a multisystem disorder named DOORS (Deafness, Onychodystrophy, Osteodystrophy, mental Retardation and Seizures). Following the first report of two recessive variants of human *TBC1D24* associated with *DFNB86* [10], six additional pathogenic variants of *TBC1D24* have been reported to associate with non-syndromic deafness (Figure 1A). Twenty-eight pathogenic variants of *TBC1D24* are associated with syndromic deafness (Figure 1A), including a novel splice-site variant c.965 + 1G > A which is segregating in a Pakistani family reported in this study (Figure 1B). At present, there are 96 variants of *TBC1D24*, including missense, nonsense, splice site, small indels and gross insertion and deletions that are associated with non-syndromic and syndromic forms of deafness as well as seizures.

There are several alternative transcripts of human *TBC1D24* including a transcript that skips micro exon 3 [10,13]. The largest *TBC1D24* transcript encodes a TBC domain (Tre-2-Bub2-Cdc16) and a TLDc domain (TBC, LysM, domain catalytic) (Figure 1A). The functions of the TBC and TLDc domains are different and pathogenic variants of TBC1D24 have been reported in both domains (Figure 1A). Some members of the large family of TBC domain-containing proteins have been shown to function as GTPase-activating proteins (GAPs), which are involved in the regulation of membrane-trafficking in partnership with Rab-GTPases [14,15]. However, the TBC domain of TBC1D24 lacks a critical arginine residue in order to function as a Rab-GTPase, an assumption that is yet to be experimentally demonstrated with purified TBC1D24 protein [10,14]. There are five proteins encoded in the human genome that have a TLDc domain and two of them, OXR1 and NCOA7, have been shown to provide an important neuro-protective role against oxidative stress [16]. It is not known whether human *TBC1D24* function in the inner ear or in the hippocampus requires the anti-oxidative function of its TLDc domain [17].

Here, we identified a novel pathogenic splice-site variant of *TBC1D24* in family PKDF1429 from Pakistan. To investigate the localization and possible function of TBC1D24 in the human inner ear, we performed immunostaining using human temporal bones. In addition, using CRISPR/Cas9 editing of *Tbc1d24,* we engineered mice with some of the same pathogenic mutations reported for human *TBC1D24*. We previously reported the first mouse model of human early infantile epileptic encephalopathy 16 (EIEE16) due to *TBC1D24* p.Ser324Thrfs*3 variant. In mouse, this variant recapitulated the seizure phenotype reported in the human family [13,18]. Here, we evaluated the phenotypes of a mouse with p.Asp70Tyr variant of *Tbc1d24* as a model of human *DFNB86* [10] and a mouse with *Tbc1d24* p.Ser178Leu variant as a model of human *DFNA65* [11,12]. We also undertook molecular modeling together with molecular dynamic (MD) simulations to evaluate the effects of a substitution of leucine for the serine-178 residue on human and mouse TBC1D24. A *Tbc1d24* p.His336Glnfs*12 allele was also engineered, which, depending on the second variant in trans-configuration, could lead to a mouse model of human DOORS or early infantile epileptic encephalopathy 16 (EIEE16) with deafness [19,20].

## 2. Materials and Methods

### 2.1. Ethic Approval and Clinical Evaluation

Our study was approved by and conducted in accordance with The Office of Human Subjects Research Protection (OH-93-N-016 to TBF) at the National Institutes of Health, Bethesda, Maryland and from the IRB-DF/2020 in Lahore, Pakistan (OH-93-DC-0016 to SR). Written informed consent was obtained from and signed by all individuals in this study. Clinical history, including a pedigree, onset of hearing loss and episodes of seizure were obtained during interviews with family members. Pure tone audiometry for all affected individuals and their parents in family PKDF1429 were obtained in home settings. The existence of morphological features of DOORS were evaluated in all affected individuals and their obligate carrier parents.

### 2.2. Whole Exome Sequencing (WES) and Bioinformatic Analyses

For family PKDF1429, sequence capture was performed using an Illumina Truseq Rapid Capture Enrichment Kit to create libraries (Illumina, San Diego, CA, USA). Genomic DNA samples of the hearing-impaired siblings III:10 and III:15 (Figure 1) were exome sequenced to an average depth of 56× and 68×, respectively. Sequence reads were mapped against the GRCh38 assembly using BWA (Burrows-Wheeler Alignment) [21] with recalibration performed using the GATK pipeline (Genome Analysis Toolkit, https://gatk.broadinstitute.org/hc/en-us). Variant calls were annotated with ANNOVAR. Variant Call Format (VCF) files were imported into Ingenuity Variant Analysis (IVA, Qiagen, Germantown, MD, USA) for further downstream analyses. Variants were prioritized based on their allele frequency, pathogenicity, conservation and zygosity. Sanger sequencing validated co-segregation of the variant with the phenotype.

### 2.3. Mouse Models of Human TBC1D24-Associated Deafness

The analyses of mice in this study were conducted according to the National Institutes of Health Guidelines for the Care and Use of Laboratory Animals. All experimental procedures were approved by the NINDS/NIDCD Animal Care and Use Committee (ACUC) at the National Institutes of Health (protocol 1263-15 to TBF and protocol NEI-626 to LD). We engineered mouse *Tbc1d24* c.208 G > T p.(Asp70Tyr) and c.[533 C > T; 534 T > G] p.(Ser178Leu) which are the equivalent of human *TBC1D24* c.208 G > T p.(Asp70Tyr) (*DFNB86*), and c.533 C > T p.(Ser178Leu) (*DFNA65*) variants, respectively [10,11,12]. In mouse, we also engineered a *Tbc1d24* c.1008delT p.(His336Glnfs*12) variant, which in human is associated with DOORS or early infantile epileptic encephalopathy 16 (EIEE16) with deafness [19,20]. Since homozygosity for the p.His336Glnfs*12 variant in mouse is an embryonic lethal, mice that are compound heterozygous for two variants p.Ser324Thrfs*3 and p.His336Glnfs*12 were studied as they are viable. *Tbc1d24* missense alleles were created using CRISPR/Cas9-mediated homologous recombination directly in C57BL/6J zygotes with a single-strand DNA oligo as the recombination template (IDT, Integrated DNA Technologies, Coralville, IA, USA) in each allele. Briefly, guide RNAs (gRNA, for SpCas9, PAM = NGG) were selected for each intended allele based on their relative positions to target codons, and ranking by the online gRNA selection tool (www.CRISPRscan.org). gRNAs were synthesized with T7 in vitro transcription as described [22] and further tested for their efficiencies of in vitro cleavage and in-cell culture indel mutagenesis activities. For the in vitro cleavage assay, genomic PCR product, containing the target sites of selected gRNAs was incubated with SpCas9 protein (NEB, New England Biolabs, Ipswich, MA, USA) by following manufacturer’s suggested protocol and analyzed on 2% agarose gel stained with ethidium bromide. Guide RNAs were further tested for their efficiencies inducing indels at target sites in an immortalized mouse embryonic fibroblast (MEF) cell line engineered to carry a tet-inducible Cas9 expression cassette. Upon confirmation of efficient target cleavage activity in MEF cells, gRNAs were mixed with SpCas9 protein (PNA Bio, Thousand Oaks, CA, USA) along with a synthetic single-strand donor DNA oligo template as described above. The single-strand donor DNA oligo templates were designed to repair the cleavage gap by the gRNA to restore the open reading frame, while having the desired single amino acid changes introduced based on Richardson et al. [23]. The mixture of gRNAs, Cas9 protein and the donor oligos were microinjected into zygotes of C57BL/6 background as described [24]. gRNAs used to generate *Tbc1d24* mutations p.Asp70Tyr, p.Ser178Leu and p.His336Glnfs*12 were 5′-GGCATAAGGTGTGACTGTG-3′, 5′-GTCATACAGGAAGACTCAA-3′ and 5′-CTGAGTGGAAGTTCTCTGCA-3′, respectively. Donor oligo sequences of p.Asp70Tyr, p.Ser178Leu and p.His336Glnfs*12 were 5′- GAGCCACACCCTGCGCGGGAAAGTGTACCAGCGCCTGATCCG GGACATCCCCTGCCGCACAGTCACACCTTATGCCAGCGTGTACAGTGACATTGT-3′ (96 mer oligo), 5′-GATGAAGCTGAGTGTTTCGAAAAAGCCTGCCGCATCTTATCCTGCAATGACCC CACCAAGAAGCTCATTGACCAGAGCTTCCTGGCCTTTGAGTTGTCCTGTATGACATTTGGGGACCTGGTGAACA-3′ (127 mer oligo) and 5′-ACTGTGGGCTCTGATCCCTCCTCGCTTTTCCCAG GCAGTTTGTGCACTTAGCTGTCCAGCAGAGAACTTCCACTCAGAGATTGTCAGCGTGAAGGA-3′ (96 mer oligo), respectively.

F0 founder mice were screened for their CRISPR/Cas9 edited mutations as described below. Mice carrying each mutation were backcrossed to C57BL/6J mice for more than three generations. The p.Asp70Tyr allele was genotyped by Sanger sequencing; a 403 bp amplified fragment was generated by PCR using primers (5′-CAGCCTAGGACCTGCCTTG-3′, 5′-TGGCAATACACAGGAGGATCT-3′). Mice carrying the p.Ser178Leu variant were genotyped using PCR primers 5′-CGCAAGATCCTCCTGTGTATTG-3′ and 5′-AGAACTTGAGGATGGCCAGA-3′. The resulting 409 bp amplicon was analyzed after a *HinFI* restriction endonuclease digestion (NEB). The wild type amplicon was cut once with restriction endonuclease *HinFI* producing 187 bp and 222 bp restriction fragments. The p.Ser178Leu amplicon was uncut by *HinFI.* p.His336Glnfs*12 mice were genotyped by PCR using primers 5′-GGGACTTCTAGGAATAATTTCACC-3′ and 5′-TGCTGCAGTGATGAGAAGAG-3′. The resulting 325 bp amplicon was analyzed after a *NlaIIII* restriction endonuclease digestion (NEB). The wild type 325 bp amplicon was digested by *NlaIIII* and produced 115 bp and 210 bp restriction fragments, while the p.His336Glnfs*12 allele generated a 324 bp amplicon and was uncut by *NlaIIII.* The engineering and genotyping of mice carrying the p.Ser324Thrfs*3 allele were described in Tona et al. 2019 [13].

The human *TBC1D24* p.Ser178Leu variant is associated with DFNA65 autosomal dominant nonsyndromic hearing loss with an onset reported in the third decade and initially affecting the high frequencies [11,12]. Since C57BL/6J mice show age-related hearing loss due to the *Cdh23* c.753G > A variant [25,26], mice carrying the *Tbc1d24* p.Ser178Leu allele were backcrossed with *Cdh23* wild type (*Cdh23*^753G^) mice with a C57BL/6J background (B6.CAST-Cdh23^Ahl+^, The Jackson Laboratory, Bar Harbor, ME). In this study, all *Tbc1d24* p.Ser178Leu mice were homozygous for *Cdh23*^753G^, which was genotyped by PCR using primers 5′-CTAGAGAACCCACGCAGGAC-3′ and 5′-TCAGCCCAAGCCTCTACTGT-3′. The resulting 430 bp amplicon was analyzed after *BsrI* restriction endonuclease digestion (NEB). The *Cdh23*^753G^ allele was uncut by *BsrI* while *Cdh23*^753G>A^ allele generated 66 bp and 364 bp restriction fragments.

### 2.4. ABR and DPOAE Measurements of Hearing Ability

Auditory testing was performed in the NIDCD/NIH Mouse Auditory Testing Core facility as described [27]. Briefly, mice were anesthetized by an intraperitoneal (IP) injection of a combination of 56 mg/kg body weight of ketamine (VetOne, MWI, Boise, ID, USA) and 0.375 mg/kg body weight of dexdomitor (Putney, Portland, ME, USA). Both auditory brainstem responses (ABR) and distortion product otoacoustic emissions (DPOAE) were measured in the right ear using Tucker-Davis Technologies hardware (TDT, Alachua, FL, RZ6 Multi I/O processor, MF-1 speakers) and software (BioSigRz, v. 5.7.2). ABR wave 1 latencies and amplitudes were measured at 80 dB SPL at 8 kHz, 16 kHz, 32 kHz and 40 kHz. To evaluate the hearing status of the *Tbc1d24* p.Asp70Tyr allele, five wild types (1 male, 4 females), 6 heterozygotes p.Asp70Tyr (1 male, 5 females) and 6 homozygotes p.Asp70Tyr (1 male, 5 females) were tested at P30, P60 and P90. For the *Tbc1d24* p.Ser178Leu allele with *Cdh23* c.753G > A variant (*Cdh23*^753G>A^), three wild types (2 males, 1 female), three heterozygotes p.Ser178Leu (2 males, 1 female) and three homozygotes p.Ser178Leu (2 males, 1 female) were tested at P30, P60 and P90. The *Tbc1d24* p.Ser178Leu was homozygous for the wild type *Cdh23* (Cdh23^753G^) and there were three wild types (2 males, 1 female), four heterozygotes p.Ser178Leu (3 males, 1 female) and six homozygotes p.Ser178Leu (4 males, 2 females), which were tested at P30, P60, P90 and P180. Since *Tbc1d24* compound heterozygous p.Ser324Thrfs*3/p.His336Glnfs*12 mice die by P20, three heterozygous p.Ser324Thrfs*3 (1 male, 2 females) and three compound heterozygous p.Ser324Thrfs*3/p.His336Glnfs*12 (1 male, 2 females) mice were evaluated only at P17.

### 2.5. Immunofluorescence Staining in Celloidin Sectionsof Human Cochleae

Human temporal bone specimens were obtained within 12 to 24 h of death from subjects without a hearing loss history [28]. The temporal bones were stored in 10% neutral-buffered formalin at 4 °C for 4 weeks, decalcified with 5% EDTA for 9 to 12 months, dehydrated in a graded ascending ethyl alcohol series and embedded in celloidin over a 3-month period. Temporal bones embedded in celloidin were cut in 20 µm thick serial sections of which every tenth section was mounted and stained with hematoxylin and eosin. The rest of the sections were stored in a glass jar and immersed in 80% ethanol and 20% double distilled water. Celloidin sections containing the cochlea (mid-modiolar area) were removed from the jar and mounted on Superfrost Plus slides (Thermo Fisher Scientific, Waltham, MA, USA) and used for immunofluorescence staining.

To remove the celloidin [28], sections were placed in a glass Petri dish and immersed in ethanol saturated with sodium hydroxide solution (100 g NaOH in 100 mL of ethanol), diluted 1:3 with 100% ethanol for 1 h, followed by 100% ethanol, 50% ethanol, and distilled water for 10 min each. Slides were placed horizontally in a glass Petri dish containing antigen retrieval solution (Vector Antigen Unmasking Solution, Vector Labs, Burlingame, CA, USA diluted 1:250 with distilled water) and heated in the microwave oven using intermittent heating of two 2 min cycles with an interval of 1 min between the heating cycles, Slides were allowed to cool for 15 min, followed by 10 min wash with phosphate buffered saline solution (PBS, 0.1 M, pH 7.4). A drop of trypsin antigen retrieval solution (#ab970, Abcam, Boston, MA, USA 1 drop of concentrated trypsin in 4 drops of PBS for 3 min) was added to each section. Sections were washed with PBS 4 × 15 min.

Sections were blocked in PBS containing 1% bovine serum albumin fraction V (Sigma-Aldrich, St. Louis, MO, USA) and 0.1% Triton X-100 (Sigma-Aldrich, St Louis, MO, USA) for 1 h, and incubated subsequently with the rabbit antibody against TBC1D24 (ab101933, RRID: AB_10712373 or ab234723, Abcam) and mouse monoclonal antibody against acetylated tubulin (T745, Sigma-Aldrich, St Louis, MO, USA) in blocking solution for 72 h at 4 °C in a humid chamber. After 4 washes (15 min) in PBS, a mixture of Alexa-488 conjugated goat anti-rabbit polyclonal IgG and Alexa-564 conjugated horse anti-mouse polyclonal IgG (both from Molecular Probes, Carlsbad, CA, USA) was added at a dilution of 1:500 in blocking solution and incubated at room temperature for 3 h in the dark. Slides were mounted in Vectashield mounting media (Vector Labs) containing DAPI. Background immunofluorescence was removed using the Vector True VIEW kit (Vector Labs).

Negative controls, consisting of secondary antibody only, and unstained human cochlea sections were used to assess for background staining and auto-fluorescence, respectively. Negative controls exhibited minimal staining or auto-fluorescence. As positive controls, mouse cochlea sections were immuno-stained with TBC1D24 antibody as previously reported [10]. The Institutional Review Board (IRB) of UCLA approved this study of human temporal bones (IRB protocol #10-001449, AI) and methods used in this study are in accordance with NIH and IRB guidelines and regulations. Informed consent was obtained from each patient before inclusion in the study of temporal bone sections; temporal bone specimens came from five individuals with normal hearing, of which three were female and two were male, ranging from 55 to 78 years old.

### 2.6. In Situ Hybridization and Immunohistochemistry Using Mouse Cochleae

In situ hybridizations were performed using RNAscope assays (Advanced Cell Diagnostics (ACD), Newark, CA, USA). Cochleae from C57BL/6J wild type mice at postnatal day three (P3) were fixed overnight at 4 °C in 4% PFA (Electron Microscopy Sciences, Hatfield, PA, USA) in 1 PBS. Fixed cochleae were cryopreserved in 15% sucrose in 1× PBS for overnight at 4 °C and then in 30% sucrose in 1× PBS for overnight at 4 °C. Each cochlea was embedded and frozen in Super Cryoembedding Medium (Section-Lab, Hiroshima, Japan). Frozen cochleae were sectioned (12 µm thick) using a CM3050S cryostat microtome (Leica, Vienna, Austria). RNAscope Multiplex Fluorescent V2 Assays (ACD) were conducted using Probe-Mm-Tbc1d24 (target region: 839 to 1739 nucleotides, NM_001163847.1), Probe-Mm-Tubb3-C4 (target region: 2 to 1636 nucleotides, NM_023279.2) and Probe-Mm-Myo7a-C2 (target region: 1365 to 2453 nucleotides, NM_001256081.1). Images were taken with an LSM880 confocal microscope equipped with 63× and 40× objectives (Carl Zeiss Microscopy, Thornwood, NY, USA).

Immunolocalization of TBC1D24 protein was examined in mouse cochleae at P8. The dissected cochleae were fixed overnight with 4% PFA in 1× PBS at 4 °C. Permeabilization was done with 0.5% Triton X-100 and blocking was performed with 2% BSA and 5% goat serum in 1× PBS. To verify localization of TBC1D24 protein, we used four antibodies (ab101933, RRID: AB_10712373, Abcam; ab234723, Abcam; NBP1-82925, RRID: AB_11061868, Novus Biologicals, Littleton, CO; sc-390237, Santa Cruz Biotechnology, Dallas, TX, USA). Samples were incubated with one of the anti-TBC1D24 antibodies and anti-Tubulin Beta 3 (TUBB3) antibody (801202, RRID: AB_10063408, BioLegend, San Diego, CA, USA) for 2 h, washed with 1× PBS and stained with anti-rabbit and anti-mouse secondary antibodies (Alexa Flour 488 and 568, respectively). Specimen were mounted using ProLong Gold Antifade Mountant with DAPI (Thermo Fisher Scientific, Carlsbad, CA, USA) and observed with the 63× objective using LSM780 (Carl Zeiss Microscopy).

### 2.7. scRNA-Seq

Single cell RNA-Seq (scRNA-Seq) datasets [29] of the developing cochlear epithelium at E16.5, P1 and P7 were analyzed. Briefly, normalized count tables of E16.5, P1 and P7 cochlear samples were obtained from GSE137299. Genes without canonical names (starting with “Gm-” or ending with “Rik”) were removed before further analyses. Modularity-based clustering with Leiden algorithm was implemented in Scanpy (v1.4.5). Briefly, principal component analysis (PCA) was performed on all remaining genes. A KNN graph was constructed based on the Euclidean distance by the function *pp.neighbors* using default settings. Cells were clustered by the function *tl.leiden* with the resolution E16.5 = 10, P1 = 2 and P7 = 0.8. Expression of *Myo6* was used as a general hair cell marker and *Fgf8* expression was used as an inner hair cell (IHC) marker. *Neurod6* was used as an outer hair cell (OHC) marker of E16 and P1 data, and *Slc26a5* was used as a P7 OHC marker. Violin plots for *Myo6*, *Myo15, Cldn11* and *Tbc1d24* at E16.5, P1 and P7 were generated by Seaborn (v0.10.1) in Python (v3.8.2).

### 2.8. Computational Modeling

A three-dimensional (3D) structure of the TBC domain of human TBC1D24 (hTBC1D24) was obtained using template-based modelling. A critical step in molecular modelling is the selection of a template. The implementation of Hidden-Markov profiles in template-search algorithms increases sensitivity, especially in the identification of templates that share very low sequence identity with the query sequence, but still share the same fold. This is the case of the fold-recognition algorithm implemented in HHpred server [30], which led to the identification of the TBC domain of Skywalker (PDB id: 5HJQ and resolution 2.3 Å) [31] as a suitable template of the human TBC domain of TBC1D24 after scanning the hTBC1D24 profile against each of the profiles of the structures deposited in the Protein Data Bank [32]. The initial alignment of hTBC1D24 and 5HJQ amino-acid sequences covers residues 11-311 of hTBC1D24 (NP_001186036.1) and shows 24% amino acid sequence identity, with all the secondary structural elements aligned. This alignment was subsequently refined using an iterative process that places the most conserved residues packing towards the core of the protein and avoids gaps within secondary-structural elements. This refinement was guided by the conservation scores calculated by the Consurf server [33] and by the ProQ2 score calculated for each residue position (local ProQ2 score) [34], which evaluates the compatibility between the TBC1D24 sequence and the structural fold at a given segment. The final alignment of hTBC1D24 (residues 11-314) and 5HJQ (residues 55-338) together with that of mouse and human TBC1D24 obtained with Clustal Omega (https://www.ebi.ac.uk/Tools/msa/clustalo/) was then used to create 2000 models of human TBC (hTBC) and mouse TBC (mTBC) domains using Modeller [35]. The final models were those with the best ProQ2 score and Procheck analysis from each set and covered more sequence than that reported by Finelli et al. [36]. α-Helical restraints were applied during the two production runs to residues 300-314 (NP_001186036.1) to complete the C-terminal helix of the TBC domain, which was missing in the template.

### 2.9. Molecular Dynamic Simulations

The all-atom models of human and mouse TBC were enclosed in a simulation box of dimensions ~97 × 97 × 120 Å comprising a hydrated bilayer and 150 mM KCl as electrolyte. Each system included ~116,000 atoms. The initial setup for a hydrated bilayer was made of 288 POPC (1-palmitoyl-2-oleoyl-sn-glycero-3-phosphocholine) and one PIP2 lipid molecules and was generated by using the CHARMM-GUI web-based interface [37] and then relaxed for 50 nanoseconds (ns). For each system, first, water molecules were placed within the protein structure with Dowser [38]. The protein was then inserted into the simulation box by superimposing the coordinates of the PIP2 heads into the lipid bilayer associated with the protein model and removing the overlapping water molecules. To optimize the protein-solvent and protein-lipid interfaces in the model, and to thermalize the system, a series of short simulations were carried out with gradually weaker positional restraints applied to the protein, the PIP2 head, the Dowser-added water molecules, and the z-coordinates of the lipid atoms (excluding hydrogen atoms). The whole relaxation process was carried out for 23 ns. In order to run two independent simulations, the relaxation was repeated starting from a different set of velocities. Finally, two fully independent unrestrained simulations were carried out for a total of 400 ns for each system.

The final simulation snapshots were extracted and used to initiate the free-energy perturbation (FEP) calculations. Here, a given side-chain, Ser178, is mutated alchemically to leucine using a step-wise protocol controlled by a parameter λ that reflects the weights of the serine and the leucine in the potential energy function of the system. FEP calculations were carried out with 21 intermediate λ steps. For each λ, the system was simulated for 50 ns. The initial 10 ns of each simulation were considered as an equilibration. The final 40 ns were split into two sets to have two independent estimates of the free-energy. Free energy differences were evaluated by using the BAR algorithm [39]. To evaluate the free-energy of the unfolded state, the FEP protocol was repeated for a tripeptide (Ala-Ser-Ala), capped with acetyl and N-methylamine groups at the N and C termini, embedded in a cubic water box of dimensions ~97 × 97 × 97 Å with KCl 150 mM as electrolyte. In this case, the snapshot after 50 ns of simulation was used to initiate the FEP calculations.

All molecular dynamics simulations were carried out with Gromacs version 2018.3 [40], using an integration time-step of 2 femtoseconds (fs), periodic boundary conditions, and a Nose-Hoover temperature coupling set to 303.15 K. The pressure was maintained at 1 bar using Parrinello-Rahman coupling semi-isotropically in the x, y plane and z direction. Electrostatic interactions were calculated using the Particle-Mesh-Ewald algorithm, with a real-space cut-off of 12 Å. A shifted Lennard-Jones potential, also cut-off at 12 Å, was used to compute van-der-Waals interactions. The CHARMM36 force field for proteins [41] and lipids [42] was used in all calculations.

## 3. Results

### 3.1. Novel Splice Variant of Human TBC1D24 Associated with Deafness and Seizures

Clinical histories and phenotypes were obtained from four affected and two unaffected siblings and their unaffected parents of family PKDF1429 (Figure 1B). All affected individuals in family PKDF1429 had congenital pre-lingual profound hearing loss (Figure 1C). Individual ΙΙΙ:9 also had simple partial seizures while III:15 had tonic-clonic seizures in childhood from six months to three years of age. Individuals ΙΙΙ:10 and III:14 did not show seizures. Physical examinations in this family did not reveal any dysmorphic features of DOORS. These data indicated that deafness segregating in this family appears to be non-syndromic (only hearing loss) for individuals ΙΙΙ:10 and III:14, although admittedly subtle seizures in these two individuals may not have been noticed.

To determine the molecular genetic causes for the apparent non-syndromic and syndromic hearing loss segregating in family PKDF1429, whole exome sequencing (WES) was performed using gDNA from individuals III:10 and III:15 (Figure 1B). Compound heterozygous variants of *TBC1D24* (NM_001199107.1): c.641G > A p.(Arg214His) and c.965 + 1G > A were identified in affected individuals III:10 and III:15, which were confirmed by Sanger sequencing (Figure 1D). The donor splice-site pathogenic variant c.965 + 1G > A of intron 2 of *TBC1D24* is novel. The c.965 + 1G > A is absent from gnomAD (v2.1.1) database of 125,748 exome and 45,708 whole genome sequences from unrelated individuals and the G nucleotide is conserved from human to lamprey with a CADD score of 35, indicating deleteriousness of this variant (https://cadd.gs.washington.edu/). The novel variant has been submitted to ClinVar (https://www.ncbi.nlm.nih.gov/clinvar/) under submission ID SUB7952908. The variant of c.641G > A p.(Arg214His) was reported in compound heterozygosity with variants p.Glu153Lys or with p.Val445Glyfs*33 and associated with non-syndromic deafness *DFNB86* [43]. Sanger sequencing confirmed the genotypes of eight members of family PKDF1429. Parents of affected children were heterozygous, confirming the trans-configuration of the two variants, and unaffected members of the family were either heterozygous for p.Arg214His variant or homozygous for the wild type allele (Figure 1B). The clinical heterogeneity of affected members of family PKDF1429 suggests that the *TBC1D24* genotype alone does not unequivocally dictate the phenotype, i.e., either non-syndromic deafness or the combination of deafness and seizures.

### 3.2. TBC1D24 Protein Localization in Human and Mouse Temporal Bone and Tbc1d24 mRNA Expression in Wild Type Mouse Cochlea

In P3 wild type mouse cochlea, in situ hybridization using RNAscope probes demonstrated that *Tbc1d24* mRNA was expressed in spiral ganglion neurons. However, no *Tbc1d24* mRNA was detected in mouse organ of Corti, including inner and outer hair cells (Figure 2A). Moreover, in RNAseq databases, there were only background levels of *Tbc1d24* mRNA expression present in mouse hair cells at E16.5, P1 and P7. It was similar to the background levels of *Cldn11* mRNA in hair cells, which we used as our negative hair cell expression control (Appendix A). *Cldn11* is abundantly expressed in the basal cells of the stria vascularis, but not in hair cells [44,45,46,47]. In addition, we immunolocalized TBC1D24 protein in the cell bodies of mouse spiral ganglion neurons (Figure 2B) corroborating published data [10,13]. However, TBC1D24 has also been reported to be localized in stereocilia at E14.5 and P0 to P3 but not at P7 using an sc-390237 antibody [12], an observation we reproduced with this same commercial antiserum against TBC1D24. Nevertheless, three other antibodies against TBC1D24 that showed immunoreactivity in spiral ganglion neurons, did not detect any immunofluorescence signal in hair cells at various ages of mouse, including adult. Among these three antibodies, the rabbit polyclonal ab101933 antibody was developed against an antigenic sequence of human TBC1D24 (aa 467–515) that is present within the antigenic sequence of human origin of the mouse monoclonal sc-390237 antibody (aa 437–559). The specificity of the second out of these three antibodies, NBP1-82925 antibody, was confirmed previously using *Tbc1d24* p.Val67Serfs*4 (TBC1D24 null) mouse inner ear [14]. By comparison, in temporal bone sections from deceased presumably normal hearing individuals, TBC1D24 protein was immunolocalized in spiral ganglion neurons and outer and inner hair cells using two different TBC1D24 antibodies from Abcam, including ab101933 antibody (Figure 3). Taken together, these data indicate that mouse hair cell TBC1D24 immunoreactivity using mouse monoclonal sc-390237 antibody is likely to be non-specific. In both human and mouse, there is abundant expression of *TBC1D24* mRNA and TBC1D24 protein in spiral ganglion neurons, but in hair cells, TBC1D24 expression is detected only in human, while it was undetectable in mouse hair cells.

### 3.3. Auditory Function in the Mouse Models of DFNB86 and DFNA65

The hearing status of mouse models of human *DFNB86* and *DFNA65* deafness was quantitatively evaluated using auditory brainstem responses (ABR) and distortion product otoacoustic emissions (DPOAE), which are measures of spiral ganglion neurons and outer hair cell functions, respectively. For the p.Asp70Tyr allele, which is a mouse model of the human DFNB86 p.Asp70Tyr variant [10], ABRs were measured at P30, P60 and P90 in heterozygous and homozygous mutant mice and control homozygous wild type littermates (Figure 4A and Appendix A). ABR thresholds were within the normal range for all three genotypes. TBC1D24 protein was localized to spiral ganglion neurons in both human and P8 mouse. Therefore, we focused on ABR wave 1 because wave 1 originates from the action potential in the auditory nerve [48]. ABR wave 1 latencies and wave 1 amplitudes were measured at 80 dB SPL at four frequencies (8, 16, 32 and 40 kHz). No significant differences were detected among the three genotypes (Figure 4A), suggesting normal auditory nerve function in *Tbc1d24* homozygous p.Asp70Tyr mutant mice. Homozygous p.Asp70Tyr mice and their littermate controls also have normal distortion product otoacoustic emissions (DPOAE), indicating that outer hair cell amplification of basilar membrane vibrations is indistinguishable from the wild type (Figure 4B).

The human *TBC1D24* c.533C > T p.(Ser178Leu) variant is associated with dominantly inherited loss of hearing DFNA65, segregating in two apparently unrelated families [11,12]. Affected individuals in these two families showed progressive high frequency hearing loss [11,12]. The Ser178 residue is conserved in human and mouse TBC1D24 (Appendix A), although there are other differences in amino acid sequence nearby the Ser178 residue and elsewhere between wild type human and mouse TBC1D24 proteins. We engineered a mouse model of the human *TBC1D24* dominant p.Ser178Leu allele associated with deafness DFNA65. Both homozygous and heterozygous p.Ser178Leu mutant mice with a C57BL/6J genetic background have hearing indistinguishable from their wild type littermates (Appendix A). Given that mice with a C57BL/6J background show age-related high frequency hearing loss due to *Cdh23* c.753G > A variant [25,26], both wild type littermates and p.Ser178Leu mutant mice showed the expected age-related hearing loss due to this allele of *Cdh23* (Cdh23^753A^) [25,26]. To exclude association of progressive hearing loss in this *Tbc1d24* p.Ser178Leu mouse with p.Ser178Leu variant, we measured ABR and DPOAE in *Tbc1d24* p.Ser178Leu mouse with the wild type *Cdh23* allele (*Cdh23*^753G^). Homozygous and heterozygous p. Ser178Leu mice with the *Cdh23*^753G^ allele in the background also showed normal ABR thresholds, wave 1 latencies, wave 1 amplitudes and DPOAE (Figure 4C,D and Appendix A), arguing that the observed age-related hearing loss was exclusively due to *Cdh23* (Cdh23^753A^) allele.

To evaluate the auditory function of a mouse model of human *TBC1D24* syndromic deafness, we tested hearing by ABR analyses using compound heterozygous p.Ser324Thrfs*3/p.His336Glnfs*12 mice. The human *TBC1D24* p.His336Glnfs*12 variant is associated with syndromic hearing loss. Compound heterozygosity for human p.His336Glnfs*12 and the 1206 + 5G > A exon 5 donor splice site variant of *TBC1D24* is associated with DOORS [19], whereas compound heterozygosity for p.His336Glnfs*12 and perhaps the less disabling p.Asp11Gly single amino acid substitution is associated with early-infantile epileptic encephalopathy 16 (EIEE16) with hearing loss [20]. Using CRISPR/Cas9 editing, we engineered the p.His336Glnfs*12 allele in mouse *Tbc1d24*. However, homozygous p.His336Glnfs*12 mice die during embryonic development. Human p.His336Glnfs*12 homozygotes have not been reported. Consequently, we generated compound heterozygous p.Ser324Thrfs*3/p.His336Glnfs*12 mice. As homozygous p.Ser324Thrfs*3 mice had severe seizures but normal hearing at two weeks of age [13], the p.Ser324Thrfs*3 allele was a logical variant to evaluate the p.His336Glnfs*12 allele in compound heterozygosity. Compound heterozygous p.Ser324Thrfs*3/p.His336Glnfs*12 mice also died around P20 due to seizures; however, their auditory functions at P17 were within the same range as their normal hearing heterozygous p.Ser324Thrfs*3 littermates (Appendix A). These findings indicate that several different variants of *Tbc1d24* in mouse, unlike the homologous human recessive and dominant variants of *TBC1D24*, are not associated with hearing loss. However, the seizure phenotypes in humans, due to variants of *TBC1D24* are recapitulated in our mouse *Tbc1d24* models [13].

Human TBC1D24 has an essential function required for normal hearing, since several different human variants of *TBC1D24* are associated with non-syndromic deafness or syndromic deafness (Figure 1A). However, in mouse, TBC1D24 appears not to be required for hearing perhaps because the loss of TBC1D24 function may be compensated by a paralogous protein. However, there might be other explanations as to why particular homozygous or compound heterozygous variants of *TBC1D24* are deafness-causing in humans, while the very same variants in mouse *Tbc1d24* did not affect hearing. Next, we used computational modeling and molecular dynamic simulations to identify additional non-mutually exclusive reasons beyond the cell type-specific differences in the expression of TBC1D24 to explain the different species-specific phenotypic outcome.

### 3.4. Template-Based Models of Mouse and Human TBC1D24

The final models of human and mouse TBC domain of TBC1D24 show just one residue in disallowed regions from the PROCHECK analyses, indicating that the models have overall good stereochemistry. In order to analyze the compatibility of the TBC1D24 sequence with the fold of the template, 5HJQ, we calculated the ProQ2 score normalized by the number of amino acid residues (global ProQ2 score). Both models have a normalized ProQ2 score of ~0.6, similar to that of the template (0.7). In summary, both analyses highlight the good stereochemical quality of the models and adequate selection of the template. Both models were used to analyze in further details the sites close to residues Asp70 and Ser178 in mouse and human TBC for any structural difference (Figure 5A). In particular, we considered the neighboring residues within 6 Å of Asp70 and Ser178. In both mouse and human, Asp70 is located in a loop and exposed to the solvent with adjacent residues that are strictly conserved (Figure 5A,B and Appendix A), indicating that this residue may be part of a post-translational motif or a partner-protein binding site. This observation suggests that deafness arising from p.Asp70Tyr substitution in human, but normal hearing in p.Asp70Tyr mouse, is not likely due to differences in the identity of the neighboring amino acids. Alternatively, the different phenotypic outcomes are related to an evolutionary divergence in the functional necessity and cell-type specific regulation of expression of human *TBC1D24* compared to mouse *Tbc1d24*.

The Ser178 residue is located in a α-helix (Figure 5A) and is part of a hydrophobic site with neighboring residues that are mostly conserved. However, there is a notable difference between mouse and human at residue 167. In mouse, there is a lysine and in human there is an arginine (Figure 5B and Appendix A). Even though both residues are positively charged, the chemical nature of the arginine provides a more bulky side chain compared to lysine and may indicate why the p.Ser178Leu substitution may result in a different hearing phenotype in mouse compared to human when expressed in the same cell types. Nevertheless, specific details at a structural level of this different phenotype cannot be identified by analyzing mouse and human TBC models (Appendix A). Therefore, we took advantage of Molecular Dynamics (MD).

To gain insight on the effect of p.Ser178Leu substitution, we carried out a series of MD simulations of human hTBC and mouse mTBC, enclosed in a simulation box comprising a hydrated bilayer. One PIP2 molecule anchored the protein to the bilayer (Appendix A). For each system, we first carried out two independent, unrestrained simulations of 200 ns. During these simulations, the protein structures conserved essentially all the structural features of the initial models, with hTBC and mTBC sharing the same tertiary structure except for a remarkable conformational difference (Figure 6A and Appendix A). In fact, the C-terminal helix of each domain, which in the initial models laid close to the membrane plane, moved toward Ser178, practically packing the serine side chain in between residues 304 and 308. This conformational change was observed in all simulations and, except in one case, it took place either during the relaxation phase or within the first 2 ns of unrestrained simulations (Figure 7). This finding suggests that the Ser178 residue is critical for packing the C-terminal helix to the protein core, and, as a consequence, stabilizing the folding of the global TBC domain.

To cast light on the role in stabilization of the TBC domain that is played by the Ser178 residue and how the substitution of leucine affects it, we carried out free-energy perturbation (FEP) simulations. Here, Ser178 is mutated, through a number of unphysical (alchemical) intermediates to leucine using a step-wise protocol controlled by a parameter λ that reflects the weights of serine and leucine in the potential energy function of the system. To relate the calculated free-energy values to the folding free-energy differences, we performed similar FEP simulations of a tripeptide in water, which is adopted as representative of the domain unfolded state. The substitution of a leucine-178 for serine-178 showed little effect on the hTBC domain. In fact, the calculated ∆∆G was 0.47 kcal/mol, therefore comparable to the thermal energy fluctuations, in the direction of a slight destabilization of the fold. On the other hand, surprisingly, the same mutation stabilized the folding of mTBC by 3.1 kcal/mol (Appendix A). Leucine is significantly more hydrophobic than serine and is found among the most frequent amino acids at α-helical interfaces of soluble proteins, whereas serine is mostly found in non-interfacial regions [49]. Therefore, it is not surprising that the p.Ser178Leu mutation is not disruptive and has a stabilizing effect, at least in mouse. However, that does not explain the observed disparity between human and mouse. As mentioned, the most notable difference among the amino acids adjacent to Ser178 in the two structures is Arg167 in human, which is Lys167 in mouse. These residues belong to the loop region from 162 to 167 and, therefore, are expected to be loosely structured. However, Figure 6B shows that murine Lys167 is strongly tethered in between Asp163 and Asp170 during the simulations, while the bulkier side chain of human Arg167 is more mobile. Consequently, the associated loop 162–167 is less rigid in human than in mouse. Therefore, we speculate that human residue 178 is more exposed to the solvent and to the electrostatic fluctuations arising from the conformational changes the neighboring charged residues (Arg166, Arg167, Asp163 and Asp170), while murine residue 178 faces a more compact pocket. This arrangement would then favor the more hydrophobic leucine over serine, in mouse more than in human. In summary, FEP simulations indicate that p.Ser178Leu has a stabilizing effect in mouse, but not in human.

## 4. Discussion

Pathogenic variants of human *TBC1D24* are associated with a spectrum of skeletal and neurological disorders including deafness, seizures, onychodystrophy, osteodystrophy and intellectual disability [5]. There is decisive data supporting an association of deafness with variants of human TBC1D24 (Figure 1A). In this study, we report recessive variants of *TBC1D24* c.641G > A p.(Arg214His) and c.965 + 1G > A segregating in a non-consanguineous Pakistani family. In affected individuals of this family, the donor splice-site mutation of exon 2 (c.965 + 1G > A) is novel and present in trans to c.641G > A p.(Arg214His). Bakhchane and colleagues reported that individuals with *TBC1D24* compound heterozygous p.Arg214His/p.Val445Glyfs*33 and p.Arg214His/p.Glu153Lys exhibited non-syndromic deafness *DFNB86* [43]. Surprisingly, in two affected members of family PKDF1429 (Figure 1) a compound heterozygous genotype resulted in deafness with seizure, while for the same compound heterozygous genotype the other two siblings in this family have non-syndromic deafness *DFNB86*. A common phenomenon for variants of *TBC1D24* is that phenotype varies depending upon the second pathogenic variant in trans. For example, compound heterozygosity for p.Glu153Lys and p.Arg214His resulted in *DFNB86* non-syndromic deafness [43], whereas p.Glu153Lys with p.Ala39Val or p.Glu153Lys with p.Thr182Serfs*6 in trans cause seizure without deafness [50,51].

The distinctly different functions of TBC and TLDc domains have been individually studied in orthologues, but not in TBC1D24, which is the only protein that has both of these domains. The TBC domain is a GTPase activator and TLDc domain is neuro-protective against oxidative stress [16,17]. Surprisingly, there is no genotype-phenotype correlation yet with variant location, either with the location in the *TBC1D24* gene or in one or the other of the two domains of the TBC1D24 protein as illustrated in Figure 1A [52]. Perhaps the pleiotropy, associated with different variants of *TBC1D24,* results from a composite of disabled binding motifs for an array of interacting protein partners of TBC1D24. To date, the reported binding partners of TBC1D24 include ARF6 (ADP-ribosylation factor) [53,54] and ephrinB2 [55].

A second non-mutually exclusive possibility to explain the great variety of phenotypes associated with variants of TBC1D24 is a variable contribution to compensation among individuals resulting from their different genetic backgrounds [56,57]. For example, the *TBC1D24*-associated phenotype may be influenced by common polymorphic variants of one or more of the 26 other TBC-containing proteins [58] or variants of one of the other four TLDc-containing proteins in the mammalian genome [16,17].

To determine if the inner ear cell-type-specific expression is the same in human and mouse, we performed immunohistochemistry on human temporal bones. TBC1D24 protein localized in human SGN just as we and others have reported in mouse SGN [10,12,13]. However, TBC1D24 protein was not reliably detected in mouse hair cells but was detected in human inner and outer hair cells of five different temporal bones (Figure 3B). These data suggest that one possible cause of deafness in human from variants of *TBC1D24* is a necessary function of human TBC1D24 in the sensory epithelium of the inner ear. In mouse, TBC1D24 is not detected in hair cells by RNAscope probes and is at a background level in RNAseq data, comparable with the level of CLDN11, which is known to be expressed in basal cells of the stria vascularis, but not in hair cells of early postnatal mouse inner ear (Appendix A). We also cannot rule out the possibility that human TBC1D24 may also have a necessary function in the spiral ganglion neurons as well.

Using CRISPR/Cas9 gene editing, we engineered in mouse the same deafness-causing variants as in human *TBC1D24*. Unexpectedly, *Tbc1d24* mutant mice have normal auditory functions even though they have biallelic recessive or dominant missense mutations orthologous to either of two variants associated with human non-syndromic deafness, *DFNB*86 and *DFNA65* (Table 1). There is precedent for animal models not recapitulating a human inherited pathology. For example, a variety of engineered mouse models of Huntington disease (HD) do not reproduce the severe constellation of neuropathology cascades, observed in HD patients or features of HD postmortem brains [59,60].

In mouse, homozygosity for the p.His336Glnfs*12 allele results in embryonic lethality, while compound heterozygosity of p.His336Glnfs*12 with p.Ser324Thrfs*3 produces postnatal death at about P20 due to seizures. Given that heterozygous p.Ser324Thrfs*3 mice do not show seizure and otherwise appear to be phenotypically wild type [13], we conclude that the phenotype of the p.His336Glnfs*12 allele in compound heterozygosity with p.Ser324Thrfs*3 is pathogenic. These results indicate that the p.His336Glnfs*12 variant in mouse is pathogenic but does not result in a hearing loss. In addition to our results, a heterozygous *Tbc1d24*^tm1b(EUCOMM)Hmgu^ mutant mouse has normal hearing [36] and in our laboratory the homozygote for *Tbc1d24*^tm1b(EUCOMM)Hmgu^ obtained from the KOMP repository at the Baylor College of Medicine is also an embryonic lethal.

Since TBC1D24 is localized in human and mouse spiral ganglion neurons (Figure 2 and Figure 3B), we evaluated ABR wave 1 latency and amplitude (Figure 4A,C). However, there was no significant difference among three genotypes. In mouse, homozygotes for *Tbc1d24* p.Asp70Tyr variant have normal hearing despite deafness associated with homozygosity for p.Asp70Tyr in human. In the mouse cochlea, *Tbc1d24* mRNA and TBC1D24 protein were detected only in the spiral ganglion neurons (Figure 2A,B). This result is different from the localization of human TBC1D24, which was detected in both organ of Corti and spiral ganglion neurons (Figure 3B). Perhaps in the mouse, expression of TBC1D24 protein in spiral ganglion neurons is not necessary for normal hearing. While TBC1D24 is not expressed in mouse hair cells, it is an open question as to whether TBC1D24 has a necessary function in human spiral ganglion neurons, hair cells or both. To answer this question, in future studies we will introduce in the presence of a homozygous null allele of the endogenous mouse *Tbc1d24* gene, a human wild type BAC transgene or a mutant *TBC1D24*. Would such mice, expressing only functional human *TBC1D24,* show detectable TBC1D24 mRNA and protein in hair cells and would the human TBC1D24 be necessary for hearing in mice? In a study of Parkinson disease (PD) in mouse models, a mutant α-synuclein encoded by human *SNCA* expressed from a human transgene resulted in a human-like PD-associated phenotype in mice homozygous for a null allele of the endogenous mouse *Snca* gene [61].

The genetic background in humans and mice may have a significant impact on variants of human *TBC1D24*, mouse *Tbc1d24,* or both. For example, there may be a modifier variant in the human genome that enhances the deafness phenotype of biallelic pathogenic variants of *TBC1D24*. We engineered *Tbc1d24* mutant mice using only a C57BL/6J background. The phenotype of a variant in a mouse gene can change substantially, depending upon genetic background [62]. We have yet to explore the possibility that the recessive variant p.Asp70Tyr and the dominant variant p.Ser178Leu in mouse may result in deafness when placed in the context of a different genetic background. The genotype-phenotype relationship of human TBC1D24 variants associated with seizures are recapitulated in variants of mouse *Tbc1d24* [13], unlike deafness. Does the genetic background of the B6 strain provide functional compensation for a disabled TBC1D24 protein? Might there be a strain-specific modifier in the background of a different mouse inbred strain that would suppress the non-penetrance of deafness in a B6 background? Compensation for the loss of TBC1D24 in mouse may be provided by a paralog in the mouse genome or compensation by a gene involved in the same network or signaling pathway [63]. For example, a dominant variant of *METTL13* (*DFNM1*) completely suppresses recessive non-syndromic deafness *DFNB26,* associated with a variant of *GAB2* with both genes functioning in the HGF/MET signaling pathway [64,65,66].

Another non-mutually exclusive possibility to explain the divergent outcomes of the same variant of human *TBC1D24* and mouse *Tbc1d24* is that some variants are damaging in human but only in combination with other wild type substitutions in TBC1D24 protein that have become fixed during human evolution. From molecular dynamic simulations, we provide data indicating that p.Ser178Leu, a dominant variant with a pathogenic leucine at residue 178, destabilizes human TBC1D24 protein; but surprisingly the same substitution stabilizes mouse TBC1D24, despite the considerable sequence identity between human and mouse TBC domains.

In summary, we report a novel pathogenic splice-site variant of *TBC1D24* segregating in a Pakistani family, and describe several mouse models of human *TBC1D24* associated with DFNB86, DFNA65 and syndromic deafness. We propose various possible explanations for the differences in phenotypes despite the same variant in mouse *Tbc1d24* and human *TBC1D24,* and provide experimental data from molecular dynamics to support one of many possible explanations for this species-specific outcome. Nevertheless, a comprehensive understanding as to how human *TBC1D24* variants cause DFNB86 deafness, as well as a panoply of other allele-associated abnormalities remains to be explored. Future studies will focus on the networks and protein complexes in which TBC1D24 functions in the auditory system and brain.

## Figures and Tables

**Figure 1 genes-11-01122-f001:**
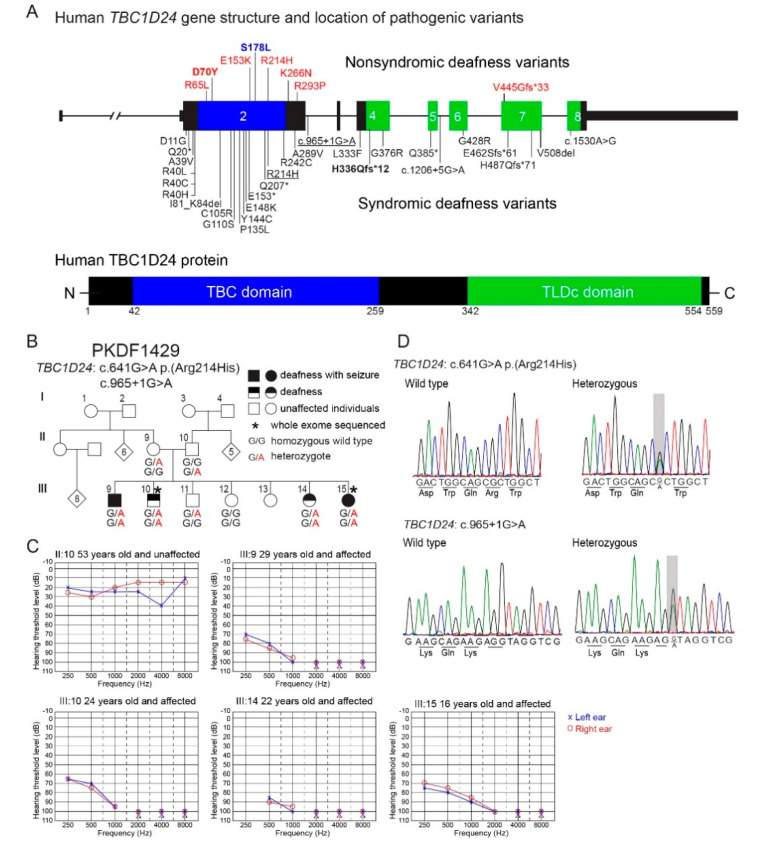
(**A**) A diagram of the human *TBC1D24* gene displaying the approximate locations of pathogenic variants associated with non-syndromic deafness and syndromic forms of deafness. The eight annotated exons of human *TBC1D24* encode a TBC domain (blue) and a TLDc domain (green). Eight reported pathogenic variants associated with non-syndromic deafness are drawn above the gene structure. p.Ser178Leu (blue) is an autosomal dominant variant associated with progressive deafness DFNA65 reported to be segregating in two unrelated families [11,12]. Other non-syndromic variants (red font) are associated with DFNB86 recessively inherited deafness. Twenty-eight pathogenic variants including three splice-site variants are associated with syndromic forms of deafness (deafness and epilepsy or DOORS) and are shown under the schematic of the gene. p.Arg214His and a splice-site variant c.965 + 1G > A (underlined) are segregating in family PKDF1429 in this study. The murine equivalent of the known human pathogenic variants of p.Asp70Tyr, p.Ser178Leu and p.His336Glnfs*12 together with p.S324Tfs*3 are characterized in this study. A, Alanine; C, Cysteine; D, Aspartic acid; E, Glutamic acid; F, Phenylalanine; G, Glycine; H, Histidine; I, Isoleucine; K, Lysine; L, Leucine; N, Asparagine; P, Proline; Q, Glutamine; R, Arginine; S, Serine; T, Threonine; V, Valine; Y, Tyrosine; *, stop codon. (**B**) Pedigree of Pakistani family PKDF1429. A novel likely pathogenic donor splice-site variant c.965 + 1G > A and a previously reported p.Arg214His pathogenic variant are associated in family PKDF1429 with apparent non-syndromic deafness and syndromic deafness, as two deaf individuals in this family were reported to have had seizures. All affected individuals self-reported to be congenitally profoundly deaf. Individual ΙΙΙ:9 (29 years old) had simple partial seizures and III:15 (16 years old) had tonic-clonic seizures in childhood from six months to three years. Individuals ΙΙΙ:10 (24 years old) and III:14 (22 years old) have not had seizures that were obvious to their parents. (**C**) Audiograms from member of family PKDF1429. All affected individuals showed bilateral profound deafness. (**D**) Representative chromatograms of genomic DNA sequences of c.641G > A p.(Arg214His) and c.965 + 1G > A with wild type allele. The variants are shaded in gray.

**Figure 2 genes-11-01122-f002:**
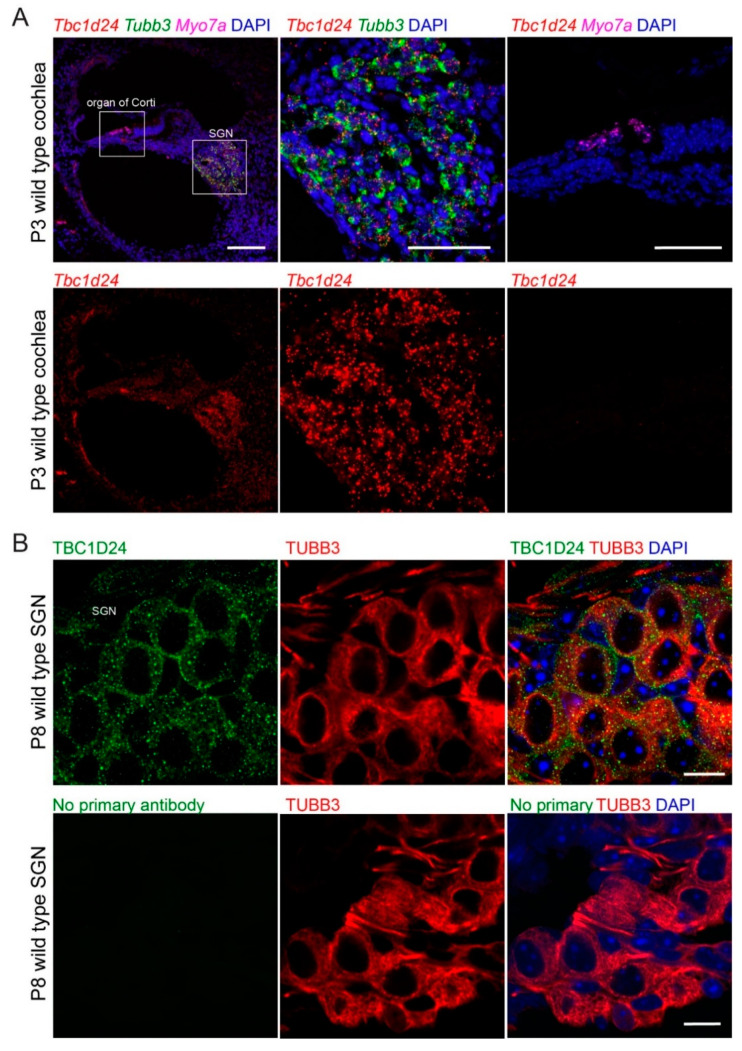
Expressions of mouse *Tbc1d24* mRNA and TBC1D24 protein in spiral ganglion neurons (SGN). (**A**) In situ hybridization using an RNAscope probe in wild type mouse cochlea at P3. *Tbc1d24* mRNA (red, probe-Mm-Tbc1d24) is present in SGN. Expression of *Tbc1d24* mRNA was not detected in the organ of Corti. Hair cells and SGN were labeled using RNAscope probes that recognize *Myo7a* (magenta, Probe-Mm-Myo7a-C2), and *Tubb3* (green, Probe-Mm-Tubb3-C4), respectively. Middle panels are enlarged images of the SGN. Right panels are enlarged images of the organ of Corti. Scale bars are 100 µm (left panel) and 50 µm (middle and right panels). (**B**) Localization of TBC1D24 in wild type mouse SGN. TBC1D24 (green) colocalizes with TUBB3 (red), a marker for the cell body of spiral ganglion neurons, in P8 mouse cochlea. Scale bars are 10 µm.

**Figure 3 genes-11-01122-f003:**
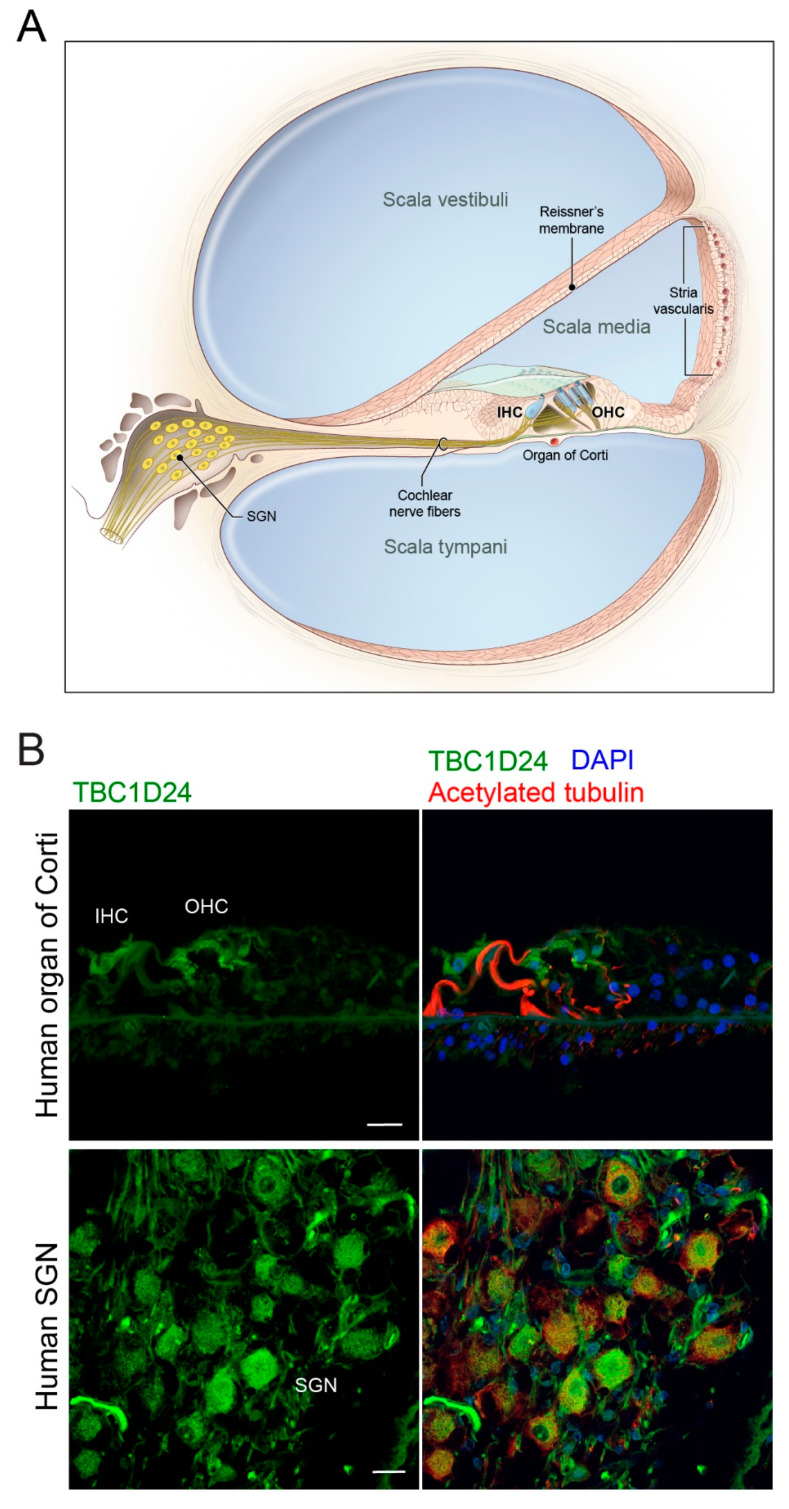
TBC1D24 expression in spiral ganglion neurons (SGN) and organ of Corti from human temporal bones. (**A**) Drawing of a cochlea in cross-section including the organ of Corti and SGN. The organ of Corti contains various supporting cells, one row of inner hair cells (IHC) and three rows of outer hair cells (OHC). (**B**) Representative images of localization of TBC1D24 in human temporal bone cross-sections. TBC1D24 is detected in an IHC and OHC (upper panels) and SGN (lower panels). Scale bars are 20 µm.

**Figure 4 genes-11-01122-f004:**
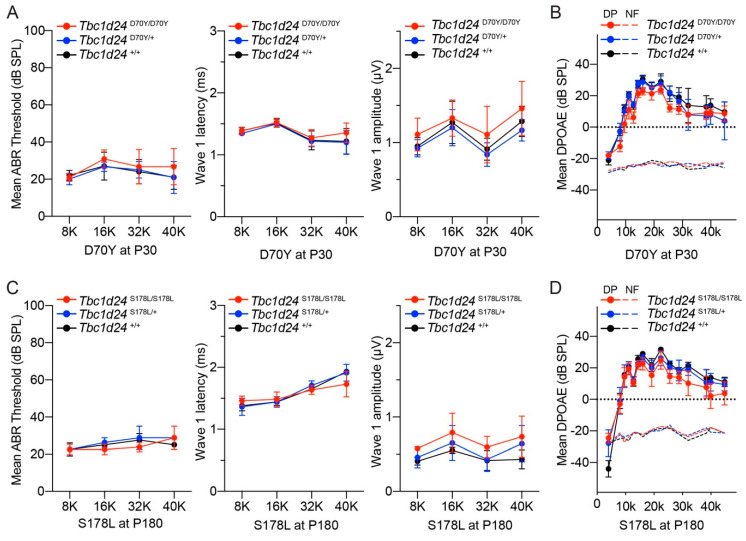
Auditory function measurements of wild type, heterozygous and homozygous *Tbc1d24* mutant mice. (**A**) Mean ABR thresholds, ABR wave 1 latency and ABR wave 1 amplitude of *Tbc1d24* homozygous p.Asp70Tyr (D70Y) (*n* = 6), heterozygous p.Asp70Tyr (*n* = 5) and wild type littermate (*n* = 5) mice at P30. No significant differences were detected between homozygous p.Asp70Tyr, heterozygous p.Asp70Tyr and wild type in threshold, wave 1 latency and wave 1 amplitude. ABR threshold at P60 and P90 are shown in Appendix A. (**B**) Mean DPOAE levels for the same *Tbc1d24* p.Asp70Tyr mice tested with ABR. There was no significant difference among the three genotypes. (**C**) Mean ABR thresholds, ABR wave 1 latency and ABR wave 1 amplitude of *Tbc1d24* homozygous p.Ser178Leu (*n* = 6), heterozygous p.Ser178Leu (S178L) (*n* = 4) and wild type littermates (*n* = 3) at P180. As *Tbc1d24* p.Ser178Leu mice and wild type littermate controls had homozygous wild type alleles of *Cdh23* (*Cdh23*^753G^), they didn’t show a high frequency progressive age-related hearing loss. Homozygous p.Ser178Leu mice and heterozygous p.Ser178Leu mice have normal hearing at P30 to P180. ABR threshold of *Tbc1d24* p.Ser178Leu with *Cdh23*^753G^ and *Tbc1d24* p.Ser178Leu with *Cdh23*^753G>A^ at P30 to P90 are shown in Appendix A. (**D**) Mean DPOAE levels in mice with the same *Tbc1d24* p.Ser178Leu allele but the wild type *Cdh23*^753G^ were tested by ABR. Dash lines indicate noise floors (NF). No significant differences were detected between homozygous p.Ser178Leu, heterozygous p.Ser178Leu and wild type controls. All data represent mean ± SD.

**Figure 5 genes-11-01122-f005:**
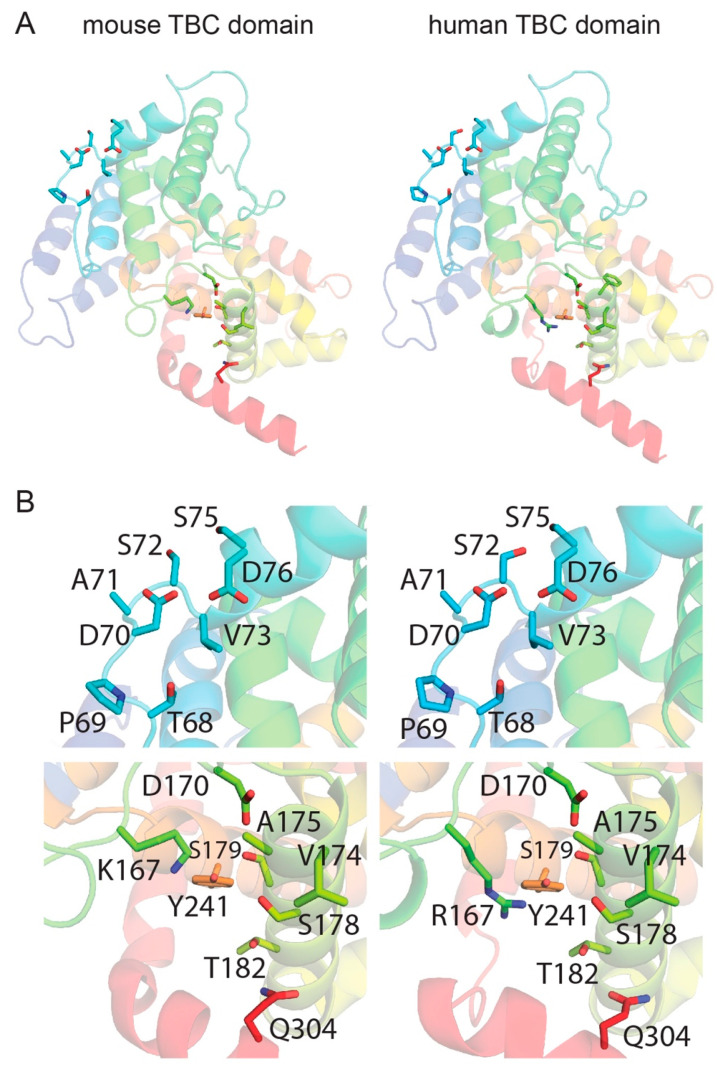
Computational modeling of human TBC1D24 and mouse TBC1D24. (**A**) Models of the TBC domains of mouse (left) and human (right) of TBC1D24 shown as a cartoon. Residues Asp70 and Ser178 as well as those within 6 Å are shown as sticks. (**B**) Enlarged view of the Asp70 (top) and Ser178 (bottom) sites in mouse (left) and human (right) models. Residues within 6 Å of Asp70 or Ser178 are also shown as sticks and indicated with labels. A, Alanine; D, Aspartic acid; K, Lysine; P, Proline; Q, Glutamine; R, Arginine; S, Serine; T, Threonine; V, Valine; Y, Tyrosine.

**Figure 6 genes-11-01122-f006:**
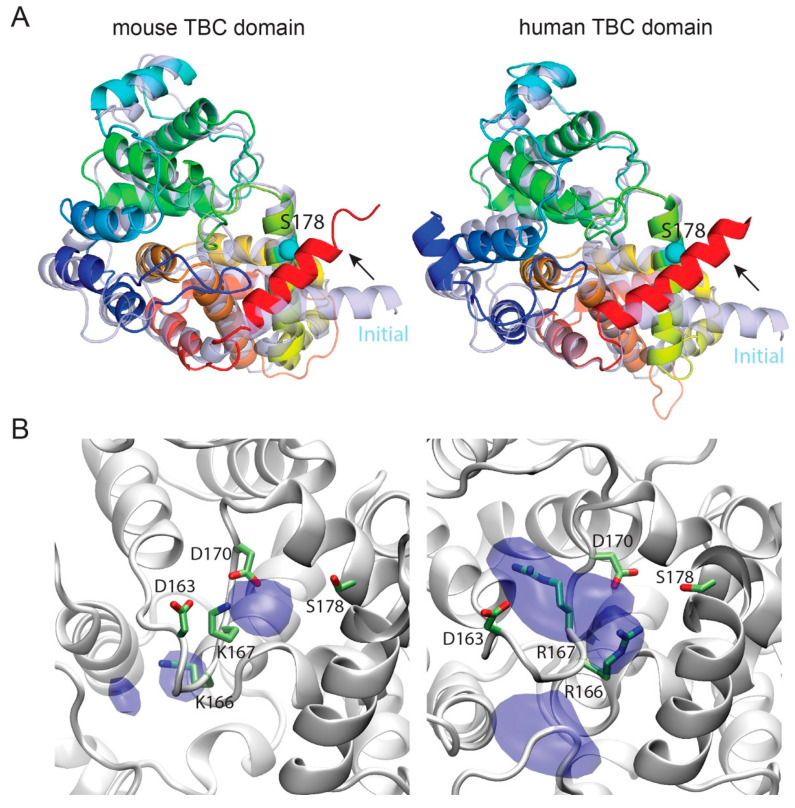
(**A**) Structural superimposition of the initial models of mouse (left) and human (right (light blue colored) with their corresponding final structures after 200-ns molecular dynamics (MD) simulations (rainbow-colored). The reorientation of the C-terminal helix of the TBC domain when comparing the initial model (α-helix colored in light red) and that after Molecular Dynamics simulation was applied (α-helix colored in bright red) is indicated as an arrow. The Cα of Ser178 in both cases is shown as a cyan sphere. (**B**) Side-chain distribution of residues 166 and 167. The isosurfaces of the distribution of the guanidinium moieties of Arg166 and Arg167 (right panel), calculated from the hTBC simulations and that of the ε-amine moieties of Lys166 and Lys167 (left panel), calculated from the mTBC simulations (nitrogens only), are shown in dark blue. In both cases the isosurfaces refer to the density value of 0.05 atomic mass units per Å^3^. The protein snapshots correspond to the final snapshots from the respective simulations (sim. 2) where the protein is shown as a gray cartoon and the side-chains of residues Asp163, Lys/Arg166, Lys/Arg167 and Asp170 as gray sticks with the headgroups colored by element: N is blue; O is red. D, Aspartic acid; K, Lysine; R, Arginine; S, Serine.

**Figure 7 genes-11-01122-f007:**
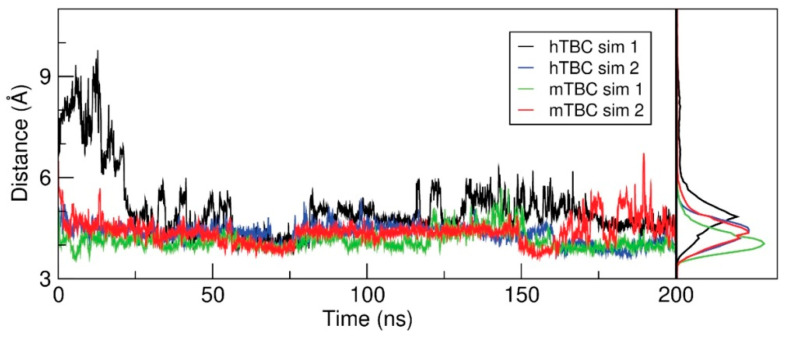
Distance between Ser178 and the C-terminal helix. Left panel: The minimal distance between the Cβ of Ser178 and the Cα of the residues from 304 to 308 is plotted as a function of simulated time, showing how Ser178 is adjacent to the TBC C-terminal helix in most of the snapshots extracted from both human and mouse simulations. Right panel: normalized distributions of the values in the left panel.

**Table 1 genes-11-01122-t001:** *TBC1D24* variants and associated phenotypes.

Variant	Genotype	Human Phenotype	Mouse Phenotype
p.Ser178Leu	p.Ser178Leu/p.Ser178Leu	not reported	normal hearing, no seizures
	p.Ser178Leu/+	progressive hearing loss [11,12]	normal hearing, no seizures
p.Asp70Tyr	p.Asp70Tyr/p.Asp70Tyr	congenital profound hearing loss [10]	normal hearing, no seizures
	p.Asp70Tyr/+	no clinical phenotype	normal hearing, no seizures
p.His336Glnfs*12	p.His336Glnfs*12/ p.His336Glnfs*12	not reported	embryonic lethality
	p.His336Glnfs*12/ p.Asp11Gly	seizures with deafness [20]	not reported
	p.His336Glnfs*12/ c.1206 + 5G > A	DOORS syndrome [20]	not reported
	p.His336Glnfs*12/ p.Ser324Thrfs*3	not reported	seizures, postnatal death ~P20, normal hearing

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
