# Peer review of "Mouse Models of Human Pathogenic Variants of TBC1D24 Associated with Non-Syndromic Deafness DFNB86 and DFNA65 and Syndromes Involving Deafness"

_genes, 2020, doi:10.3390/genes11101122_

Round 1

Reviewer 1 Report

In this paper, the authors initially discovered a novel splice site variation in TBC1D24 in a consanguineous family segregating a phenotype of deafness and seizures as a compound heterozygote with a published missense variation. The phenotypes emphasize the pleiotropy of TBC1D24 even among members of the same family. The authors discuss different genetic backgrounds or potential very subtle neurological phenotypes that are not clinically apparent in the 2 patients classified as non-syndromic. Continuing their excellent previous work on determining TBC1D24 genotype-phenotype relationships, the authors here have generated 3 specific mouse models of DFNB86, DFNA65 and DOORS/infantile epileptic encephalopathy 16 (EIEE16) with deafness. Surprisingly, none of the mice generated had a deafness phenotype. This raises important questions regarding targeted knock-in mouse models for human HL, particularly for autosomal dominant forms of HL that largely cannot be addressed with knock-out models. The authors have also performed RNA and protein expression analysis on human and mouse material. On a protein model, the authors have generated interesting and novel molecular dynamic (MD) simulation results. It would have been helpful to have access to the supplementary material submitted during the review.

Major points

Line 83 ‘There are several alternative transcripts of human TBC1D24’.

This sentence should be rewritten to reflect on the fact that, Uniprot lists only one isoform in humans involving skipping of exon 3 and deletion of 6 residues (p.322-327) with 4 potential other isoforms where the evidence is not so strong. In the mouse, Uniprot Q3UUG6-2 also lists a second isoform (p.322-327) also with 2 potential other isoforms. This exon has been targeted by the authors in a previous paper.

By using S324Tfs*3 with H336Qfs*12, are not the authors specifically testing the effects of wild type isoform 2 (del exon 3) with H336Qfs*12? Since this generates a phenotype of ‘seizures, postnatal death ~P20’ these interesting results should be discussed with reference to the authors’ previous publication, where they concluded S324Tfs*3 is a recessive allele. Would it be expected, for example, that wild type isoform 2 could rescue a heterozygous fs such as H336Qfs*12 present in both the major and minor isoform?

The authors have generated a model of TBC1D24 with the program Modeller. However, a model of TBC1D24 has been generated with the same program described in a paper cited in the manuscript (‘Finelli et al, The epilepsy-associated protein TBC1D24 is required for normal development, survival and vesicle trafficking in mammalian neurons’). As the supplementary data file are unavailable to this reviewer-it is difficult to comment on the quality of the model. The molecular dynamic simulations described are however new and highly interesting. Supplementary Figure 6 should therefore be moved from the supplementary section to the main body of the manuscript to replace Figure 5, in case of format constraints.

Lines 292-303 (Materials and methods section 2.7. scRNA-SeqLines) and 394-5 ‘Moreover, in RNAseq databases, there were only background levels of Tbc1d24 mRNA expression present in mouse hair cells at E16.5, P1 and P7.’ The authors state that (some) of the differences found between human and mouse may be due to different functional spectrums of TBC1D24 in both species with overlapping functions in neurons in both species but critical function in hair cells only in humans. This is supported by the finding that TBC1D24 was not detected in mouse cochlear hair cells but was expressed in human cochlear samples. How do the authors explain the fact that in a previous study (Azaiez et al. 2014; DOI: 10.1002/humu.22557, Fig 2) Tbc1d24 has been found to be expressed in both spiral ganglion neurons and inner and outer hair cells? The authors should use public databases such as https://umgear.org/ or https://shield.hms.harvard.edu/or to compare their expression data.

Mouse models have been without any doubt a tremendous asset in the study of human hearing loss. One would generally believe that fine-tuning of mouse models of human disease with CRISPR technique (targeted knock-in) would allow an even more accurate reflection of the human phenotypes then traditional KO models. At least for TBC1D24 and the presented models here this seems not to be the case. This raises important questions regarding targeted knock-in mouse models for human HL, particularly so for autosomal dominant forms of HL that largely could not be addressed previously with KO models, including the TBC1D24 S178L variant. Do the authors think this could have implications for the future design and use of targeted mouse models or would the conclusions be specific to TBC1D24? What could be a possible strategy to evaluate the suitability of mouse models for human HL beforehand to avoid the large expense related to the creation of models? These questions would be interesting to be briefly addressed in the discussion.

Minor points.

The manuscript should be carefully edited throughout.

Please list all the mouse models in the abstract and final paragraph of the introduction.

Remove the designation PKDF1429 throughout.

In Fig 1D, replace chromatogram c.641G>A if available-the heterozygous change is not visible at the resolution supplied.

Legend to figure 1 ‘Variants D70Y, S178L and H336Qfs*12 were each individually 117 introduced into mouse Tb1d24 and are characterized in this study’ is confusing as other fs variants were also analyzed. This sentence should be rewritten ‘The murine equivalent of the known human pathogenic variants D70Y, S178L and H336Qfs*12 together with S324Tfs*3…..’

Replace SGN for SGNs throughout the manuscript. SGNs are defined twice in the Fig 2 legend.

‘Expressions of mouse Tbc1d24 mRNA and TBC1D24 protein in spiral ganglion neurons’ should be rewritten ‘Expression of mouse Tbc1d24 mRNA and TBC1D24 protein in spiral ganglion neurons (SGN)’.

Specify that Fig 2b is an IHC in the legend.

In Fig 3a remove the ‘s’ from OHCs in the diagram and legend and the ‘s’ from IHCs in the legend.

List the gmomAD version used.

How was the seizure phenotype evaluated in mice?

The methods text does not state the age at which the DFNA65 model mice were tested. Only figure 4C shows it (p180).

Please add a reference to the materials and methods on how S324Tfs*3 was generated.

Line 545 ‘and TBC mTBC’ delete TBC.

If H336Qfs*12/D11G and H336Qfs*12/c.1206+5G>A are not reported, they should be removed from table 1.

Move the sentence lines 643-645 ‘We previously reported the first mouse model of human TBC1D24 S324Tfs*3 associated with early infantile epileptic encephalopathy 16 (EIEE16), and in mouse this variant recapitulated the seizure phenotype reported in the human family [22,56].’ To the end section of the introduction.

Author Response

Reviewer #1

Comments and Suggestions for Authors

“In this paper, the authors initially discovered a novel splice site variation in TBC1D24 in a consanguineous family segregating a phenotype of deafness and seizures as a compound heterozygote with a published missense variation. The phenotypes emphasize the pleiotropy of TBC1D24 even among members of the same family. The authors discuss different genetic backgrounds or potential very subtle neurological phenotypes that are not clinically apparent in the 2 patients classified as non-syndromic. Continuing their excellent previous work on determining TBC1D24 genotype-phenotype relationships, the authors here have generated 3 specific mouse models of DFNB86, DFNA65 and DOORS/infantile epileptic encephalopathy 16 (EIEE16) with deafness. Surprisingly, none of the mice generated had a deafness phenotype. This raises important questions regarding targeted knock-in mouse models for human HL, particularly for autosomal dominant forms of HL that largely cannot be addressed with knock-out models. The authors have also performed RNA and protein expression analysis on human and mouse material. On a protein model, the authors have generated interesting and novel molecular dynamic (MD) simulation results. It would have been helpful to have access to the supplementary material submitted during the review.

Major points

Line 83 ‘There are several alternative transcripts of human TBC1D24’.

This sentence should be rewritten to reflect on the fact that, Uniprot lists only one isoform in humans involving skipping of exon 3 and deletion of 6 residues (p.322-327) with 4 potential other isoforms where the evidence is not so strong. In the mouse, Uniprot Q3UUG6-2 also lists a second isoform (p.322-327) also with 2 potential other isoforms. This exon has been targeted by the authors in a previous paper.”

 Response: We know experimentally and have published that there are several alternative transcripts encoded by the human TBC1D24 and mouse Tbc1d24 genes. Numerous alternative transcripts of Tbc1d24 expressed in the mouse inner ear were published by us in Rehman et al., 2014, (PMID: 24387994) in Supplemental Data Figure S1. The functions and expression patterns of alternative transcripts of mouse Tbc1d24 and human TBC1D24 will be pursued in a future study. In our original text we mentioned the alternative transcripts of the human TBC1D24 and mouse Tbc1d24 containing micro exon 3 (Tona et al., 2019). In the revised manuscript on line #101 we have added the additional reference to alternative transcripts of TBC1D24/Tbc1d24 that we described in Rehman et al., 2014, which is reference #10 in the original manuscript.

“By using S324Tfs*3 with H336Qfs*12, are not the authors specifically testing the effects of wild type isoform 2 (del exon 3) with H336Qfs*12? Since this generates a phenotype of ‘seizures, postnatal death ~P20’ these interesting results should be discussed with reference to the authors’ previous publication, where they concluded S324Tfs*3 is a recessive allele. Would it be expected, for example, that wild type isoform 2 could rescue a heterozygous fs such as H336Qfs*12 present in both the major and minor isoform?”

 Response: We think the reviewer is asking if the wild type transcript from the S324Tfs*3 allele that skips exon 3 might rescue the compound heterozygote S324Tfs*3 with H336Qfs*12. This is a most interesting question for which we have an answer. In Tona et al., 2019, we show that in the wild type, transcripts with and without exon 3 are both expressed at P0. However, by P15-P20 there is very low or a barely detectable amount of the transcript that skips micro-exon 3. Please see Figure 5 in Tona et al. 2019. The majority of transcripts, qualitatively and quantitatively, in the adult wild type include micro-exon 3. You can imagine that an oligo that causes skipping of exon 3 might be therapeutic in mouse and human homozygous for the S324Tfs*3 variant, an idea that we are presently testing in collaboration with Dr. Michelle Hastings.

“The authors have generated a model of TBC1D24 with the program Modeller. However, a model of TBC1D24 has been generated with the same program described in a paper cited in the manuscript (‘Finelli et al, The epilepsy-associated protein TBC1D24 is required for normal development, survival and vesicle trafficking in mammalian neurons’). As the supplementary data file are unavailable to this reviewer-it is difficult to comment on the quality of the model. The molecular dynamic simulations described are however new and highly interesting. Supplementary Figure 6 should therefore be moved from the supplementary section to the main body of the manuscript to replace Figure 5, in case of format constraints.”

Response: 

Thank you for your insightful comments and suggestions to move Figure S6 to the main body. Regarding the supplementary file, the authors originally submitted a supplementary file to GENES together with the manuscript. We regret that the reviewer did not have access to this file. We requested and obtained approval from the GENES editorial office that we retain the original 6 figures in the main text and add the additional figure for a total of 7 figures in the main text.

Modeller is a program that uses as minimal inputs a protein template (preferably an X-ray or Cryo-EM structure) and the sequence alignment between the template and the protein to be modeled. The quality of the final model depends mostly upon two factors: 1) the resolution of the reported structure and 2) the percentage of sequence identity extracted from the alignment. As indicated by the reviewer, Finelli et al indeed used Modeller, the same program. But their analyses involved less sophisticated modeling procedures than those we used in our study. We don’t think it’s helpful to emphasize this in the manuscript since the model in Finelli et al. is very nice work, too. We have added the following phrase to the original sentence below on line 433 of the revised manuscript and have referenced Finelli et al, 2019.

The final models were those with the best ProQ2 score and Procheck analysis from each set and covered more sequence than that reported by Finelli et al, 2019. 

As indicated in the Methods, the alignment between the TBC domain from Skywalker (used as a template in our study) and hTBC1D24 was obtained through a protocol in which Hidden-Markov profiles were used as implemented in the HHpred server instead of Clustal Omega method used by Finelli et al, known to produce alignments with more gaps. The HHpred protocol increases by 40% the sensitivity in the identification of templates that are not closely related homologs and by 30% the accuracy of the initial sequence alignment as compared to other methods, such as Clustal Omega. In addition, in this work we used a state-of-the-art iterative refinement procedure to improve the initial alignment and the model. Also, we used ProQ2 and conservation scores, as stated in the Methods, to evaluate and quantify such improvement in each step. The final model obtained in our work has a ProQ2 score value similar to those of X-ray structures indicating the high quality of our model. In addition, our model extends until residue 314, compared to ~260 residues, as inferred from Fig 1 in Finelli et al. We stated this in the original manuscript. However, a direct comparison between the two models is not possible, because no model or sequence alignment was provided by Finelli et al. Finally, our strategy to relax the initial model also involved more sophisticated MD simulations than in Finelli et al. In our case, TBC is not only inserted in a simulation box with solvent and counterions, but it is also bound to a PIP2 lipid molecule embedded in a lipid bilayer. Therefore, our simulation system represents a more natural environment for TBC1D24.

“Lines 292-303 (Materials and methods section 2.7. scRNA-SeqLines) and 394-5 ‘Moreover, in RNAseq databases, there were only background levels of Tbc1d24 mRNA expression present in mouse hair cells at E16.5, P1 and P7.’ The authors state that (some) of the differences found between human and mouse may be due to different functional spectrums of TBC1D24 in both species with overlapping functions in neurons in both species but critical function in hair cells only in humans. This is supported by the finding that TBC1D24 was not detected in mouse cochlear hair cells but was expressed in human cochlear samples. How do the authors explain the fact that in a previous study (Azaiez et al. 2014; DOI: 10.1002/humu.22557, Fig 2) Tbc1d24 has been found to be expressed in both spiral ganglion neurons and inner and outer hair cells? The authors should use public databases such as https://umgear.org/ or https://shield.hms.harvard.edu/or to compare their expression data.”

Response: As the reviewer #1 pointed out, TBC1D24 has been reported to be localized in stereocilia at E14.5 and P0 to P3, but not at P7 using an sc-390237 antibody (Azaiez et al.). To evaluate the localization of TBC1D24, we used four different commercial antibodies including sc-390237. We reproduced this observation with the sc-390237 antiserum against TBC1D24. However, three other antibodies against TBC1D24 did not detect any immunofluorescence signal in hair cells at various ages of mouse. Among these three antibodies, the ab101933 antibody was developed against an antigenic sequence of human TBC1D24 (aa 467-515) that is present within the antigenic sequence of human origin of the mouse monoclonal sc-390237 antibody (aa 437-559). To validate these antibodies, we transfected pEGFP-Mm-TBC1D24 into HeLa cells with Lipofectamine 3000. After fixation, cells were immunostained using these TBC1D24 antibodies. The attached figure shows transfected HeLa cells using pEGFP-Mm-TBC1D24 and pEGFP-vector as a negative control, which were stained with ab101933 antibody, phalloidin and Hoechst. Moreover, in temporal bone sections from deceased presumably normal hearing individuals, TBC1D24 protein was immunolocalized in spiral ganglion neurons and outer and inner hair cells using two different TBC1D24 antibodies from Abcam, including ab101933 antibody, which did not stain mouse hair cells. Taken together, these data suggest that mouse hair cell immunoreactivity using mouse monoclonal sc-390237 antibody is likely to be non-specific.

To support our observation, we used published RNAseq data to show that there is no detectable expression of Tbc1d24 in mouse inner and outer hair cells. This figure was generated from data available on gEAR (see below).

“Mouse models have been without any doubt a tremendous asset in the study of human hearing loss. One would generally believe that fine-tuning of mouse models of human disease with CRISPR technique (targeted knock-in) would allow an even more accurate reflection of the human phenotypes then traditional KO models. At least for TBC1D24 and the presented models here this seems not to be the case. This raises important questions regarding targeted knock-in mouse models for human HL, particularly so for autosomal dominant forms of HL that largely could not be addressed previously with KO models, including the TBC1D24 S178L variant. Do the authors think this could have implications for the future design and use of targeted mouse models or would the conclusions be specific to TBC1D24? What could be a possible strategy to evaluate the suitability of mouse models for human HL beforehand to avoid the large expense related to the creation of models? These questions would be interesting to be briefly addressed in the discussion.”

Response: There are several examples of mouse models that do not recapitulate the human phenotype. Thankfully, the majority of mouse models faithfully exhibit the same or similar human disorder. The reasons for a different phenotypic outcome in mouse and human are most interesting and important research topics. Studies of genetic and environmental modifiers are likely to yield idea about potential therapies for human disorders.

Minor points.

“The manuscript should be carefully edited throughout.”

Response: We have checked and edited the entire manuscript.

“Please list all the mouse models in the abstract and final paragraph of the introduction.”

Response: We added the mouse models to the abstract and the final paragraph of the introduction.

“Remove the designation PKDF1429 throughout.”

Response:  A rationale for removing the family designation was not provided by the reviewer. The family designation “PKDF1429” reveals no confidential information about this family. We respectfully request that the family number designation remain in the manuscript. Many years later, we have received questions about families in old papers we have published and would prefer to retain the family designation for clarity and to avoid future mix-ups.

“In Fig 1D, replace chromatogram c.641G>A if available-the heterozygous change is not visible at the resolution supplied.”

Response: We agree and have replaced panel 1D of figure in the revised manuscript.

Legend to figure 1 ‘Variants D70Y, S178L and H336Qfs*12 were each individually introduced into mouse Tb1d24 and are characterized in this study’ is confusing as other fs variants were also analyzed. This sentence should be rewritten ‘The murine equivalent of the known human pathogenic variants D70Y, S178L and H336Qfs*12 together with S324Tfs*3…..’

Response: We edited this sentence as requested.

“Replace SGN for SGNs throughout the manuscript. SGNs are defined twice in the Fig 2 legend.”

Response: We replaced SGNs with SGN in the text, in figure legends and in the lower left panel of Figure 2.

‘Expressions of mouse Tbc1d24 mRNA and TBC1D24 protein in spiral ganglion neurons’ should be rewritten ‘Expression of mouse Tbc1d24 mRNA and TBC1D24 protein in spiral ganglion neurons (SGN)’.

Response: We agree and have edited the sentence on lines 544-555 as requested.

“Specify that Fig 2b is an IHC in the legend.”

Response: We think the reviewer meant to say Figure 3b. We have modified the sentence in the Fig 3b legend according to the reviewer’s suggestion, and it now reads” TBC1D24 is detected in an IHC and OHC (upper panels) and SGN (lower panels)”.

“In Fig 3a remove the ‘s’ from OHCs in the diagram and legend and the ‘s’ from IHCs in the legend.”

Response: We removed ‘s’ from “IHCs” and OHCs.

“List the gmomAD version used.”

Response: gnomAD v2.1.1 was used. This information was added to the main text file on line 491.

“How was the seizure phenotype evaluated in mice?”

Response: Seizures were evaluated in mouse using behavioral tests that we previously described in great detail in Tona et al, 2019. Compound heterozygous S324Tfs*3/H336Qfs*12 mice show wild running, which is a major feature of seizures in rodents. In Tona et al., 2019 there is a drawing in figure 2 that describes the various seizure manifestations in mouse that we observed (and reported by many others). We also video-recorded and then quantified the seizures in homozygous S324Tfs*3 mice compared to phenotypically heterozygous and wild type littermates.

“The methods text does not state the age at which the DFNA65 model mice were tested. Only figure 4C shows it (p180).”

Response: On page 6, line 311 we did mention the age.

“Please add a reference to the materials and methods on how S324Tfs*3 was generated.”

Response: We mentioned how we generated the S324Tfs*3 on line 266.

“Line 545 ‘and TBC mTBC’ delete TBC.”

Response: We deleted TBC as requested.

“If H336Qfs*12/D11G and H336Qfs*12/c.1206+5G>A are not reported, they should be removed from table 1.”

Response: The pathogenic variant of H336Qfs*12/D11G and H336Qfs*12/c.1206+5G>A are reported in human (Strazisar et al.2015 and Campeau et al. 2014). However, there are no reports of mouse H336Qfs*12/D11G or H336Qfs*12/c.1206+5G>A. Instead of H336Qfs*12/D11G and H336Qfs*12/c.1206+5G>A variants, we evaluated H336Qfs*12/H336Qfs*12 and H336Qfs*12/S324Tfs*3.

“Move the sentence lines 643-645 ‘We previously reported the first mouse model of human TBC1D24 S324Tfs*3 associated with early infantile epileptic encephalopathy 16 (EIEE16), and in mouse this variant recapitulated the seizure phenotype reported in the human family [22,56].’ To the end section of the introduction.”

 Response: As requested, the sentence was moved further along in the Introduction section.

Reviewer 2 Report

In this manuscript, the authors have identified a novel splice-site variant in compound heterozygosity with a missense mutation in TBC1D24 gene. Both variants segregated with the hearing loss in a Pakistani family. Taking this data as the start point, and in order to understand why mutations in this gene lead different phenotypes, they have generated several mice model (knock-in) using CRISPR/cas9. These animals exhibited normal hearing. The authors postulate several hypotheses that could explain the differences in phenotype between mouse and human. They also show experimental data based on molecular dynamics to support these differences.

The manuscript is quite complete and could be published with only minor considerations that I show below:

1-The mutation nomenclature should be the same format throughout the text, for example, p.Arg214His. I mean, they should use ever the three letters nomenclature.

2-Figure 1, Letter A), should not be in bold.

3- Authors should indicate if their analysis lets identified CNV.

Author Response

Reviewer #2

 Comments and Suggestions for Authors

“In this manuscript, the authors have identified a novel splice-site variant in compound heterozygosity with a missense mutation in TBC1D24 gene. Both variants segregated with the hearing loss in a Pakistani family. Taking this data as the start point, and in order to understand why mutations in this gene lead different phenotypes, they have generated several mice model (knock-in) using CRISPR/cas9. These animals exhibited normal hearing. The authors postulate several hypotheses that could explain the differences in phenotype between mouse and human. They also show experimental data based on molecular dynamics to support these differences.

The manuscript is quite complete and could be published with only minor considerations that I show below:

“The mutation nomenclature should be the same format throughout the text, for example, p.Arg214His. I mean, they should use ever the three letters nomenclature.”

Response: All variants were changed to the three letters nomenclature as requested.

“Figure 1, Letter A), should not be in bold.”

 Response: We changed it so that the letter is not in bold.

“Authors should indicate if their analysis lets identified CNV.”

Response: Whole exome data can potentially be used to identifying CNV. However, Hong et al. 2016 (https://doi.org/10.1186/s13073-016-0336-6) point out some of the limitations and the lack of reproducibility. We have had limited success in calling the CNVs from WES data. A number of CNV calls from WES have turned out be false positives. Recently, we have had success using MLPA analyses to identify CNV in genes of interest.

Reviewer 3 Report

It is an interesting study underlining the phenotypic heterogeneity of TBC1D24 genetic variants and providing a novel TBC1D24 donor splice-site variant. After performing a series of carefully planned in-depth experiments the authors came to the conclusion that human TBC1D24 and mouse tbc1d24 have a different expression pattern in the ear and that their mouse models do not recapitulate a human pathology, a rarely observed phenomenon that makes the studies on the role of TBC1D24 in human ear even more challenging.

Minor comments

  1. Could the different phenotypic presentation in one family (hearing loss vs. hearing loss + seizures) result from the percentage and/or the nature of mutant transcript generated by the c.965+1G>A variant?
  2. If the studied family was consanguineous, this should be indicated on the family pedigree.
  3. As WES was performed for two individuals from the family, weren’t there other genetic variants that could explain the different phenotypic presentation?
  4. Are there other “TBC1D24 families” in the literature, in which the phenotype was different for particular family members?
  5. For reporting human pathogenic variants use the guidelines provided by the Human Genome Variation Society.
  6. Line 2: delete the word “Title”.
  7. Line 293: what does “sc” in front of “scRNA-Seq” stands for?
  8. Line 545: delete one “TBC”
  9. Line 581: delete “of”
  10. Line 711: delete “[references]”

Author Response

Reviewer #3

Comments and Suggestions for Authors

“It is an interesting study underlining the phenotypic heterogeneity of TBC1D24 genetic variants and providing a novel TBC1D24 donor splice-site variant. After performing a series of carefully planned in-depth experiments the authors came to the conclusion that human TBC1D24 and mouse tbc1d24 have a different expression pattern in the ear and that their mouse models do not recapitulate a human pathology, a rarely observed phenomenon that makes the studies on the role of TBC1D24 in human ear even more challenging.”

Minor comments

  1. “Could the different phenotypic presentation in one family (hearing loss vs. hearing loss + seizures) result from the percentage and/or the nature of mutant transcript generated by the c.965+1G>A variant?”

Response: A very interesting idea but not one we can test in the family members. There is already considerable speculation in our manuscript. We would prefer to not add more.

  1. If the studied family was consanguineous, this should be indicated on the family pedigree.

Response: PKDF1429 is not a consanguineous family and thus the reason no consanguineous marriages are indicated in the Figure 1B pedigree.

We corrected the sentence on page 19 line 981 as suggested.

  1. “As WES was performed for two individuals from the family, weren’t there other genetic variants that could explain the different phenotypic presentation?”

Response: Our analysis of the WES data was focused on identifying biallelic pathogenic variants of genes that might explain the phenotype and was consistent with an autosomal recessive mode of inheritance in a non-consanguineous family. We evaluated our exome data to identify variants that could explain genetic heterogeneity in family PKDF1429 but didn’t identify variants that meet pathogenicity criteria and a segregation pattern for the disorder. We also checked our exome data for pathogenic variants in the published TBC1D24 protein partners and genes functioning in the signaling pathways relevant to TBC1D24 but didn’t identify any candidate variants.”

  1. “Are there other “TBC1D24 families” in the literature, in which the phenotype was different for particular family members?”

Response:  Banuelos et al. 2017 (PMID: 28663785) reported that two affected showed epilepsy and one affected showed epilepsy, parkinsonism, psychosis and intellectual disability in the same family.

  1. “For reporting human pathogenic variants use the guidelines provided by the Human Genome Variation Society.”

Response: We changed all variants so that they conform to the guidelines of the Human Genome Variation Society.

  1. Line 2: delete the word “Title”.

Response: “Title” was deleted.

  1. “Line 293: what does “sc” in front of “scRNA-Seq” stands for?”

Response: “sc” stands for “single cell” and it is mentioned in the text on line 399.

  1. Line 545: delete one “TBC”

Response: “TBC” was deleted.

  1. Line 581: delete “of”

Response: ‘of’ on line 581 was deleted.

  1. Line 711: delete “[references]”

Response: It was deleted.